# EXTRACTING POST-TREATMENT COVARIATES FOR HETEROGENEOUS TREATMENT EFFECT ESTIMATION

## ABSTRACT

The exploration of causal relationships between treatments and outcomes, and the estimating causal effects from observational data, have garnered considerable interest in the scientific community recently. However, traditional causal inference methods implicitly assume that all covariates are measured prior to treatment assignment, while in many real-world scenarios, some covariates are affected by the treatment and collected post-treatment. In this paper, we demonstrate how ignoring or mishandling post-treatment covariates can lead to biased estimates of treatment effects, referred to as the "post-treatment bias" problem. We discuss the possible cases in which post-treatment bias may appear and the negative impact it can have on causal effect estimation. To address the challenge, we propose a novel variable decomposition approach to account for post-treatment covariates and eliminate post-treatment bias, based on a newly proposed causal graph for post-treatment causal inference analyses. Extensive experiments on synthetic, semi-synthetic, and real-world data demonstrate the superiority of our proposed method over state-of-the-art models for heterogeneous treatment effect estimation.

## 1 INTRODUCTION

The estimation of treatment effects plays a pivotal role in decision-making across several influential domains, such as epidemiology (Dechartres et al., 2013), economics (Lin & Ye, 2007), and social sciences (Heckman, 1991). It enables the identification of causal relationships between treatment, such as smoking, and outcome of interest, for instance, heart disease. In recent years, a plethora of methods (Liu et al., 2020; Rosenbaum, 1987; Rosenbaum & Rubin, 1983; Wu et al., 2022; Frangakis & Rubin, 2002; Hullsiek & Louis, 2002; Athey & Imbens, 2016; Chipman et al., 2010; Wager & Athey, 2018; Atan et al., 2018; Hassanpour & Greiner, 2019; Johansson et al., 2016; Shalit et al., 2017; Yao et al., 2018) have emerged with a primary focus on addressing the estimation of causal effects using observational data. Nevertheless, these methods primarily center on mitigating the confounding bias introduced by confounders within the observational data through statistical or spatial mapping techniques. Other formidable challenges have not received adequate attention and resolution, just as we are about to discuss in this work, where we will explore and address the estimation bias stemming from post-treatment variables Holland (1986); Pearl (2015).

Despite the considerable success of the current methods in estimating treatment effects, they implicitly assume that all covariates are measured before the treatment or intervention is imposed and their values and distributions are not affected by the intervention, known as *pre-treatment variables* (Yao et al., 2021), and they mainly focus on eliminating the confounding bias. However, in many real-world scenarios (e.g., medical health), a significant proportion of covariates will be affected by the intervention, which are referred to as ***post-treatment variables*** (Holland, 1986; Pearl, 2015). For instance, in a study investigating the effect of smoking on the incidence of heart disease, post-treatment variables could be the occurrence of side effects (e.g., headache) or a certain medical measurement (e.g., blood pressure). Practitioners have increasingly directed their attention toward the role of post-treatment variables in causal inference. For instance, in (Bareinboim & Pearl, 2012; Bareinboim & Tian, 2015; Correa et al., 2018; Bareinboim et al., 2022), researchers utilized post-treatment variables to recover unbiased causal effects from selection bias. (Zhang et al., 2020) utilized post-treatment variables to remove confounding bias in image classification task. In this work, we focus on another problem: ignoring or mishandling post-treatment variables can lead to ***post-treatment bias*** (Montgomery et al., 2018; Coppock, 2019).

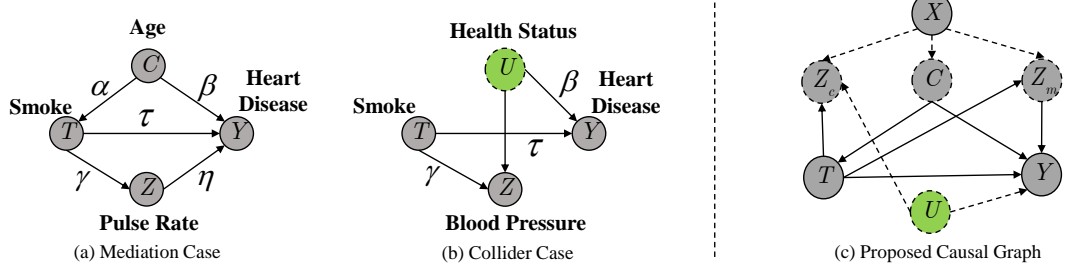

Figure 1: (a)-(b) illustrate two cases of post-treatment bias. (c) shows the proposed causal graph with observed covariates ($X$), unmeasured variables ($U$), treatment ($T$), and outcome ($Y$). $C$, $Z_m$, and $Z_c$ denote confounders, mediation, and collider post-treatment variables, respectively.

For example, as presented in Figure 1, the treatment variable $T$ indicates whether a person smokes, the post-treatment variable $Z$ represents whether a person's blood pressure or pulse rate is normal or not, the outcome variable $Y$ represents the incidence of heart disease, $C$ is the confounder (e.g., Age), and $U$ is the risk factor that affects $Z$ and $Y$, For the sake of clarity, we treat $U$ as an unobserved variable (e.g., Health Status) here. We will elaborate on other scenarios in subsequent sections of this paper. In Figure 1.(a), the causal effect of $T$ on $Y$ not only includes the direct effect from $T$ to $Y$, but also involves a mediating effect caused by the post-treatment variable $Z$. Noting that in this work we focus on the total treatment effect, which is more practical in real-world scenarios. If we fail to identify and separate $Z$ from other covariates, such as confounders $C$, when we adjust for confounders, the treatment effect through the mediation pathway can be lost, leading to biased treatment effect estimation. In Figure 1.(b), the post-treatment variable $Z$ is a collider affected by both treatment $T$ and risk factor $U$. If we condition on blood pressure $Z$ equaling normal, the treated group, i.e., the smoking population, may consist of more people with better health status than the control group. This creates an unblocked path between $T$ and $U$ in the causal graph, introducing another post-treatment bias due to the imbalance of health status between different groups resulting from conditioning on $Z$. Although recent studies (Kuroki, 2000; Pearl, 2015; VanderWeele, 2009) have discussed the harm caused by ignoring post-treatment variables in causal inference, they either address only experimental studies (Coppock, 2019; Homola et al., 2020; King, 2010; Montgomery et al., 2018) or consider post-treatment bias due to mediators alone (Li et al., 2022), ignoring other situations that may lead to post-treatment bias.

In this study, we tackle the challenge of post-treatment bias mitigation by employing representation learning techniques to derive post-treatment variables from observed covariates. We delve into the examination of two distinct scenarios capable of inducing post-treatment bias and introduce a comprehensive framework namely *PoNet*. Within this framework, we focus on inferring representations of confounding factors and post-treatment variables directly from the observed covariates. Subsequently, we put forth an inference policy designed to facilitate the estimation of heterogeneous treatment effects, aiming to achieve estimates of interest with addressing post-treatment bias.

## 2 PRELIMINARIES

### 2.1 POST-TREATMENT BIAS

We begin with the concepts of post-treatment variables that can result in post-treatment bias. As shown in Figure 1, **Mediation Post-treatment Variable**, denoted by $Z_m$, refers to the variables that is affected by the treatment $T$ and influences the outcome $Y$; **Collider Post-treatment Variable**, denoted by $Z_c$, refers to the variables that is affected by both treatment $T$ and risk factor $U$ but has no direct effect on outcome.

Incorrect handling of the two aforementioned variables can lead to post-treatment bias. To illustrate this bias, we use the linear structural causal model as an example, demonstrating the consequences of ignoring or mishandling each of the two post-treatment variables.

For the case of mediation post-treatment variable in Figure 1.(a), assuming the causal model is formulated as $Y = \tau T + \beta C + \eta Z = (\tau + \eta \gamma) T + \beta C$, the total treatment effect of $T$ on $Y$ is $\tau + \eta \gamma$, then the estimated average treatment effect from observational data can be formulated as:

$$\begin{aligned}
\Delta_a &= E(Y|T=1) - E(Y|T=0) \\
&= \tau + \beta(E(C|T=1) - E(C|T=0)) + \eta(E(Z|T=1) - E(Z|T=0)),
\end{aligned} \quad (1)$$

it is well known that eliminating confounding bias is an essential step in causal inference, and the common practice is to adjust for the confounder. However, if the mediation post-treatment variable $Z$ is not extracted and separated from confounders, the post-treatment variable $Z$ will be incorrectly adjusted (i.e., $E(Z|T = 1) - E(Z|T = 0)$ tends to be 0) when we adjust for the confounders, then the estimated average treatment effect of $T$ on $Y$ would be biased that $\Delta_a = \tau \neq \tau + \eta\gamma$, which is the mediation post-treatment bias.

For the collider post-treatment variables in Figure 1.(b), assuming the causal model is $Y = \tau T + \beta U$ where $U$ is the unmeasured variable that affects the post-treatment variable $Z$ and outcome $Y$. In this model, noting that $T$ and $U$ are independent and the causal effect of $T$ on $Y$ is $\tau$. Similarly, the estimated average treatment effect from observational data in this model can be formulated as:

$$\begin{aligned}
\Delta_b &= E(Y|T = 1) - E(Y|T = 0) \\
&= \tau + \beta(E(U|T = 1) - E(U|T = 0)),
\end{aligned} \tag{2}$$

in this model, if we condition on the post-treatment variable $Z$, a backdoor path will be open between $T$ and $U$, which means $T$ and $U$ are no longer independent, then $E(U|T = 1) - E(U|T = 0)$ in the last equation is not equal to 0 if there is an imbalance in the distribution of $U$ between the control and treated group. Therefore, the estimated treatment effect of $T$ on $Y$ is biased that $\Delta_b = \tau + \beta c \neq \tau$ where $c$ represents the discrepancy in distributions of $U$ between two groups. The detailed derivation of equation (1) and (2) can be found in Appendix.

## 2.2 CAUSAL MECHANISM FOR POST-TREATMENT MODELING

**Causal Effect Identification**. We propose a new causal graph in Figure 1 .(c) to account for post-treatment bias. Let $X$, $T$ and $Y$ denote the observed covariates, treatment and outcome, respectively. $Z_c$, $Z_m$, $C$ and $U$ represent the collider, mediation post-treatment variables, confounders and unmeasured risk factor. Here, we provide a formal theorem about the identification of heterogeneous treatment effects:

**Theorem 1.** *(Identifiability of Heterogeneous Treatment Effect) If we can recover, $p(Z_m|T, X)$ and $p(C|X)$ from the observational data, then we can recover and identify the intervention distribution for estimating heterogeneous treatment effect of $T$ on $Y$, which can be expressed by:*

$$p(Y_u|do(T), X) = \iint_{C, Z_m} p(Y_u|T, Z_m, C)p(C|X)p(Z_m|T, X), \tag{3}$$

where $Y_u$ represents the observed outcome with underlying risk factor $U = u$. It is noteworthy that $U$ and $T$ are marginally independent. Therefore, neglecting to account for the presence of $U$ in our estimates does not introduce bias in our assumption. This theorem indicates that the probability distribution of an outcome under an intervention $T$ is determined by the distribution of confounders and mediation post-treatment variables, rather than by collider post-treatment variables. This aligns directly with our analysis in Section 2.1. Proof of the theorem can be found in the Appendix.

**Minimally Sufficient Guarantee**: Building upon Theorem 1, which establishes that the treatment effect can be identified through the recovery of confounders ($C$) and mediation post-treatment variables ($Z_m$), we propose a further theorem that asserts that these variables are minimally sufficient (Silvey, 2017), for the optimal parameters $\theta$ for estimating unbiased treatment effects.

**Theorem 2.** *The joint set of inferred factors for $C$, $Z_m$ is minimally sufficient for the optimal parameters $\theta$ which estimation of unbiased treatment effects needs.*

This theorem implies that the inferred factors for $C$ and $Z_m$ encapsulate all the necessary information that is required for optimally estimating the parameters $\theta$ for the recovery of the treatment effects. A detailed proof of this theorem is provided in the Appendix.

$C$, $Z_m$ **and even $Y$ could be the risk factor**. In our earlier analysis, we elucidated the bias-creation process resulting from unmeasured risk factors. It is only natural to ponder the following questions: (1) What happens if $C$ exerts a causal influence on $Z_c$? (2) What if $Z_m$ causally affects $Z_c$? (3) What if outcome $Y$ affects $Z_c$?

For case (1), conditioning on $Z_c$ introduces a new pathway between $T$ and $Y$ through confounder $C$. The resulting bias in this case is analogous to confounding bias due to coincidence resulting

from the distribution imbalance of confounder $C$. This bias can be mitigated by adjusting for the confounders. For case (2), a similar situation arises. Conditioning on $Z_c$ introduces an additional unblocked pathway between $T$ and $Y$ through $Z_m$, and the distribution imbalance on $Z_m$ becomes the source of bias. However, unlike confounder $C$, we cannot adjust or balance $Z_m$ in the same way, as doing so would falsely erase the mediating treatment effect from $T$ to $Y$. For case (3) as mentioned in (Hernan & Robins, 2020), conditioning on collider $Z_c$ will create a new causal path between $T$ and $Y$, which will disrupt the true causal effect of $T$ on $Y$. In summary, the critical point lies in isolating $Z_c$ from the observed covariates and excluding it during inference. It is worth noting that the model and inference policy we propose in the following sections are capable of addressing both of these scenarios as long as we can recover $Z_c$ from observed covariates. Therefore, we omit these two causal pathways in the proposed causal graph, given the analysis provided above.

## 3 METHODOLOGY

### 3.1 REPRESENTATION LEARNING FOR THE THREE UNDERLYING FACTORS

**Learning of post-treatment variables**. we employ neural networks to infer the representations of post-treatment variables, $Z_m$ and $Z_c$ from observed covariates. Given the distinct nature of post-treatment variables under different treatment assignments, we construct two separate neural network channels to infer these representations. To be more precise, we seek to learn two representation functions, $f_{me}(\boldsymbol{x}, t)$ and $f_{co}(\boldsymbol{x}, t)$, with respect to the treatment assignment, mapping the observed covariates $\mathcal{X} \in \mathbb{R}^d$ to an $m$-dimensional latent space. Each treatment assignment is accommodated by parametrizing these representation learning functions through the stacking of multiple fully connected layers, resulting in representations $\boldsymbol{z}_m$ and $\boldsymbol{z}_c$ for the mediation and collider post-treatment variables, respectively.

**Learning of confounders**. Analogously, we establish a mapping function $f_c(\boldsymbol{x})$, also with $\mathcal{X} \in \mathbb{R}^d \to \mathbb{R}^m$, to derive representations of confounders from the observed covariates. This function is parametrized using multiple fully connected layers, and the resultant confounder representation is denoted as $\boldsymbol{c}$.

**Balancing confounders by optimal transport theory**. To control the confounding bias, we need to balance the distribution of the inferred representation of confounders between treated and control groups. Optimal transport theory (Villani et al., 2009; Torres et al., 2021) is a mathematical framework that allows us to measure the distance between two probability distributions. Here, we adopt the Wasserstein distance (Villani & Villani, 2009) and minimize it between the treated and control group in terms of representations of confounders. We denote the distance as $L_{wass}$ and feed it into the loss function for optimization. More details can be found in the Appendix.

### 3.2 RECONSTRUCTION MODULE

The causal graph shows that the post-treatment variable $Z_c$ is affected only by treatment and observed covariates, and has no direct impact on outcome $Y$. However, the supervised information of the training model only comes from factual outcomes in most cases, thus the lack of supervised information on $Z_c$ in training data makes it challenging to learn its representations.

To model confounders and post-treatment variables more effectively, we propose a neural network-based reconstruction module. This module incorporates learned representations of confounders, collider and mediation post-treatment variables to generate an output that closely resembles the original covariates. The reconstruction module can be formulated as:

$$\hat{\boldsymbol{x}} = \Psi(\boldsymbol{z}_m, \boldsymbol{z}_c, \boldsymbol{c}), \tag{4}$$

where $\hat{\boldsymbol{x}}$ is the reconstructed covariates, $\Psi$ is a decoder function which is parameterized by multiple fully connected layers.

### 3.3 MUTUAL INFORMATION REGULARIZER BY KERNEL DENSITY ESTIMATION

Separating confounders and post-treatment variables is essential for unbiased treatment effect estimation. When confounders' representation includes information from mediation post-treatment

variables $Z_m$, controlling for confounders may introduce mediation post-treatment bias. If $Z_m$ contains confounder information, addressing confounding bias might not be fully effective. Moreover, if $Z_m$ contains collider post-treatment information, conditioning on $Z_m$ can lead to collider post-treatment bias. Precise differentiation between confounders and post-treatment variables is thus critical for reliable treatment effect estimation.

To achieve the goal of separating confounders and post-treatment variables, we design a Mutual Information Minimization Regularizer (MIMR) based on the following corollary yielded from the causal graph in Figure 1 (c):

**Corollary 1.** *Given covariates $X$ and $T$, the confounders $C$, mediation post-treatment variables $Z_m$ and collider post-treatment variables $Z_c$ are independent to each other, i.e., $C \perp Z_m \perp Z_c || X, T$.*

Sepecifically, We propose to utilize kernel density estimation (Terrell & Scott, 1992), a non-parametric method, to fit the distributions of the representations of these variables and measure their independence. Here we take the kernel density estimation of the representations of $C$ and $Z_m$ as an example. Let $\{c^0, c^1 ..., c^N\}$ be the representation samples of confounders $C$ drawn from the marginal distribution $D_C(c)$, $\{z_m^0, z_m^1 ..., z_m^N\}$ be the representation samples of mediation post-treatment variables $Z_m$ drawn from the marginal distribution $D_{Z_m}(z_m)$, the kernel density estimates of the marginal distribution $D_C(\cdot)$, $D_{Z_m}(\cdot)$ and the joint distribution $D_{CZ_m}(\cdot)$ are given by:

$$\hat{D}_C(c) = \frac{1}{N} \sum_{i=1}^N K_h(c - c^i) = \frac{1}{Nh} \sum_{i=1}^N K(\frac{c - c^i}{h}),$$

$$\hat{D}_{Z_m}(z_m) = \frac{1}{N} \sum_{i=1}^N K_h(z_m - z_m^i) = \frac{1}{Nh} \sum_{i=1}^N K(\frac{z_m - z_m^i}{h}),$$

$$\hat{D}_{CZ_m}(c, z_m) = \frac{1}{N} \sum_{i=1}^N K_h((c - c^i)||(z_m - z_m^i)) = \frac{1}{Nh} \sum_{i=1}^N K(\frac{(c - c^i)||(z_m - z_m^i)}{h})$$

where $K(\cdot)$ is the kernel function, $h$ is the bandwidth parameter that controls the smoothness of the estimate, $||$ denotes the concatenation, and $K_n(\cdot)$ is called the scaled kernel. The kernel function can be any non-negative function that integrates to 1. In this work, we adopt the Gaussian kernel as the kernel function. Then the mutual information between $Z_m$ and $C$ can be estimated by:

$$\hat{I}(Z_m, C) = \sum_c \sum_{z_m} \hat{D}_{CZ_m}(c, z_m) log \frac{\hat{D}_{CZ_m}(c, z_m)}{\hat{D}_C(c)\hat{D}_{Z_m}(z_m)}, \tag{6}$$

similarly, we can obtain the estimated mutual information $\hat{I}(Z_c, C)$ and $\hat{I}(Z_m, Z_c)$.

## 3.4 Objective Function and Inference Policy

**Loss for predicting potential outcomes**. With the inferred representations of confounders, mediation post-treatment variables $c, z_m$ and the treatment assignment $t \in \{0, 1\}$, we can develop a prediction function $\hat{y}_i^{t_i} = f_y(c^i, z_m^i, t_i)$ parameterized by stacking fully connected layers, then minimize the mean square error (MSE) $\mathcal{L}_y = \frac{1}{N} \sum_{i=1}^N (\hat{y}_i^{t_i} - y_i)^2$.

**Loss for covariate reconstruction**. A loss function is typically defined to measure the discrepancy between the reconstructed covariate $\hat{x}$ and the true covariate $x$. One common loss function is the mean squared error (MSE), which is given by $\mathcal{L}_{re} = \frac{1}{N} \sum_{i=1}^N ||x_i - \hat{x}_i||_2^2$. Other loss functions, such as the binary cross-entropy (De Boer et al., 2005) or Kullback-Leibler divergence(Joyce, 2011), can also be used depending on the nature of the input data and the modeling objective.

**Loss for mutual information minimization regularizer**. Here we combine the three terms of estimated mutual information by kernel density estimation to guarantee the independence of confounders, mediation post-treatment variables and collider post-treatment variables from each other: $\mathcal{L}_{MIMR} = \hat{I}(Z_m, C) + \hat{I}(Z_c, C) + \hat{I}(Z_m, Z_c)$.

**Overall loss function**. The overall objective function of *PoNet* is defined by:

$$\mathcal{L} = \mathcal{L}_y + \alpha \mathcal{L}_{MIMR} + \beta \mathcal{L}_{re} + \gamma \mathcal{L}_{wass} + \eta ||\Theta||_2^2, \tag{7}$$

where $\alpha, \beta, \gamma, \eta$ are hyper-parameters to control the trade-off of the corresponding terms with the other terms, $||\Theta||_2^2$ is imposed on the learning weights $\Theta$ of the model to avoid the over-fitting.

**Inference Policy**. Based on the previous analysis, to avoid the post-treatment bias in treatment effect estimation, the inference policy should be subject to the following two rules: First, do not condition on collider post-treatment variables $Z_c$; Second, condition on the mediation post-treatment variables $Z_m$ but do not adjust them when conducting the inference.

## 4 EXPERIMENTS

### 4.1 EXPERIMENT SETTING

**Baselines**. We compare our proposed mode[1] with several baselines, which can fall into three categories: **(1) Linear regression based models**, including *OLS1* (Shalit et al., 2017): An S-learner using linear regression, treating the treatment variable as just another covariate, *OLS2* (Shalit et al., 2017): A T-learner that trains separate linear regression models for treated and control individuals; **(2) Tree based models**, including *BART* (Chipman et al., 2010): a nonparametric Bayesian regression approach based on multiple tree models, *Causal Forest* (Wager & Athey, 2018): a extension of random forest model for estimating treatment effects in causal perspective; **(3) Neural network-based models**, including *Counterfactual Regression (CFR)* (Shalit et al., 2017): A deep learning-based estimator that balances the distribution of confounders' representations, *TARNet* (Johansson et al., 2016): A variant of CFR that removes the built-in representation balancing component, *GAN-ITE* (Yoon et al., 2018): Uses Generative Adversarial Nets to capture uncertainty in counterfactual distributions and estimate the treatment effect, *CEVAE* (Louizos et al., 2017): A deep latent variable model that leverages VAE (Kingma & Welling, 2013) and proxy learning to estimate the causal effect, *TEDVAE* (Zhang et al., 2021): A variational inference approach that infers latent factors from observed variables and disentangles them for treatment effect estimation.

**Evaluation Metrics**. In this work, we adopt two widely used metrics for evaluating the performance of causal estimators. First, we adopt Rooted Precision in Estimation of Heterogeneous Effect ($\sqrt{\epsilon_{PEHE}}$) to measure the accuracy of conditional average treatment effect (CATE): $\sqrt{\epsilon_{PEHE}} = \sqrt{\frac{1}{N}\sum_{i=1}(\tau_i - \hat{\tau}_i)^2}$ where $\tau_i = y_i^{t_i=1} - y_i^{t_i=0}$ and $\hat{\tau}_i = \hat{y}_i^{t_i=1} - \hat{y}_i^{t_i=0}$ are the ground truth CATE and the estimated CATE, respectively. Second, we also adopt the mean square error (MSE) for measuring the accuracy of predicting outcomes: $\epsilon_{MSE} = \frac{1}{N}\sum_i(\hat{y}_i - y_i)^2$ in some experiments. Please refer to the Appendix for a more detailed description of the experiment setting.

### 4.2 SYNTHETIC DATA

First, we evaluate the proposed model on synthetic data. Here we only introduce the outline of the synthetic dataset due to the page limit, more details about the data generation process can be found in the Appendix. Roughly speaking, we generate the confounders (denoted by $\boldsymbol{x}_C$) with the dimension $d_C$ from a multivariate Gaussian distribution, then generate the treatment $T$ from the Bernoulli distribution based on the generated confounders. After getting the treatment assignment, we generate the mediation and collider post-treatment variables (denoted by $\boldsymbol{x}_{Z_m}$ and $\boldsymbol{x}_{Z_c}$) based on the generated treatment with the dimension $d_{Z_m}$ and $d_{Z_c}$, respectively. Then we combine the three generated factors $\{\boldsymbol{x}_C, \boldsymbol{x}_{Z_m}, \boldsymbol{x}_{Z_c}\}$ as the covariates $\boldsymbol{x}$.

**Capability of identifying each underlying factor**. Here we want to verify if the proposed model *PoNet* can identify the three underlying factors $\{\boldsymbol{x}_C, \boldsymbol{x}_{Z_m}, \boldsymbol{x}_{Z_c}\}$ from the observed covariates $\boldsymbol{x}$. In the proposed model, we develop three networks $f_c(\cdot), f_{me}(\cdot)$ and $f_{co}(\cdot)$ for learning the representations of factors of confounders, mediation and collider post-treatment variables respectively. Taking the representation learning network $f_{me}(\cdot)$ for learning $Z_m$ as an example, the first layer's dimension of the learned weights of network $f_{me}$ is $(d_C + d_{Z_m} + d_{Z_c}) \times K$ where $K$ is the dimension of the hidden layer. We can partition the learned weight matrix into two slices: (1) $S_{Z_m}$ with dimension $d_{Z_m} \times K$, that connects the variables belonging to $\boldsymbol{x}_{Z_m}$ to the representation network $f_{me}$, (2) $S_{other}$ with dimension $(d_C + d_{Z_c}) \times K$, that connects other variables not belonging to $\boldsymbol{x}_{Z_m}$ to the representation network $f_{me}$. For the network $f_{me}(\cdot)$ which can identify the mediation post-treatment variables, it is expected that the network can filter out the information of confounders $C$ and collider post-treatment variables $Z_c$, and retain the information of mediation post-treatment

---

[1]The anonymous link of the source code of the proposed model *PoNet* is: `https://anonymous.4open.science/r/Ponet-37F2/`

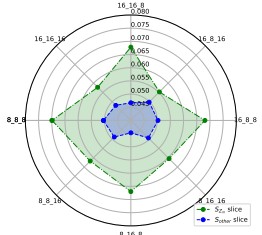 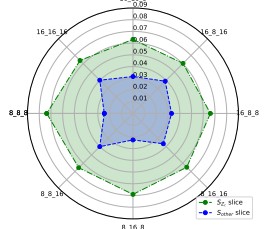 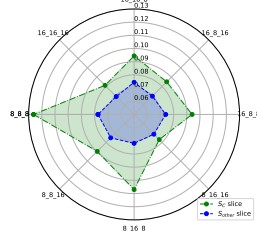

(a) Mediation post-treatment $Z_m$     (b) Collider post-treatment $Z_c$     (c) Confounders $C$

Figure 2: Radar charts visualizing the capability of PoNet in identifying the three underlying factors. Each chart represents a different underlying factor, with the vertices on the circles corresponding to different dimension settings of $\{\boldsymbol{x}_C, \boldsymbol{x}_{Z_m}, \boldsymbol{x}_{Z_c}\}$. The average absolute value of the learned weights for each dimension setting is represented by each vertex of the polygon.

variables $Z_m$. In other words, if the network $f_{me}$ can accurately identify $\boldsymbol{x}_{Z_m}$, the neuron links connected to $\boldsymbol{x}_{Z_m}$ are more active than those connected to $\boldsymbol{x}_C$ and $\boldsymbol{x}_{Z_c}$, which can be reflected in the values of the learned weights, i.e., the average absolute values of $S_{Z_m}$ is higher than that of $S_{other}$.

As shown in Figure 2, we plot the radar charts to visualize the capability of *PoNet* in identifying the three underlying factors. Each vertex on the circles represents the dimension of $\{\boldsymbol{x}_C, \boldsymbol{x}_{Z_m}, \boldsymbol{x}_{Z_c}\}$, each vertex of the polygon measures the learned weights' the average absolute value for each dimension setting. We can see that for each underlying factor, the average absolute value of $S_*$ ($* = Z_m, Z_c$ or $C$) is higher than that in $S_{other}$, which is consistent with what we expect. Therefore, it empirically shows that the proposed model is capable of identifying different underlying factors.

### 4.3 SEMI-SYNTHETIC DATA

We then evaluate the proposed model *PoNet* using the semi-synthetic dataset PeerRead (Kang et al., 2018). The Peer-Read dataset comprises peer reviews of computer science papers, with each entry representing an author. The features of each entry are bag-of-word representations extracted from

| Methods | $d = 50$ | $d = 100$ | $d = 200$ |
|---|---|---|---|
| *OLS1* | $2.241 \pm 0.481$ | $3.052 \pm 1.013$ | $3.193 \pm 1.944$ |
| *OLS2* | $2.002 \pm 0.396$ | $2.742 \pm 0.751$ | $3.023 \pm 1.354$ |
| *BART* | $2.258 \pm 0.498$ | $2.915 \pm 0.998$ | $3.385 \pm 1.958$ |
| *Causal Forest* | $2.088 \pm 0.440$ | $2.609 \pm 0.846$ | $2.973 \pm 1.743$ |
| *CEVAE* | $2.303 \pm 0.196$ | $3.037 \pm 0.340$ | $3.188 \pm 0.731$ |
| *GANITE* | $2.414 \pm 0.290$ | $2.756 \pm 0.422$ | $2.529 \pm 1.100$ |
| *TEDVAE* | $2.150 \pm 0.737$ | $2.669 \pm 0.809$ | $2.602 \pm 1.678$ |
| *TARNet* | $2.420 \pm 0.288$ | $2.582 \pm 0.720$ | $2.722 \pm 1.072$ |
| *CFR* | $2.437 \pm 0.287$ | $2.611 \pm 0.733$ | $2.720 \pm 1.098$ |
| **PoNet** | $\mathbf{1.393 \pm 0.178}$ | $\mathbf{1.869 \pm 0.502}$ | $\mathbf{2.053 \pm 0.829}$ |

Table 1: $\sqrt{\epsilon_{PEHE}}$ performance comparison on PeerRead, lower is better, $d$ denotes the dimension of post-treatment variables.

the titles and abstracts of their papers. In this dataset, each author is categorized based on whether their papers contain specific keywords, and the outcome variable is the number of citations their papers receive. To simulate the necessary variables, we generate confounders $C$ and treatment assignments and introduce artificial mediation post-treatment variables and collider post-treatment variables $Z_m$ and $Z_c$, respectively, based on the generated treatments. For more information on the detailed data generation process, please refer to the Appendix.

**Treatment effect estimation**: Here we consider the different dimension of the post-treatment variables as $d = 50, 100, 200$ and evaluate the performance of *PoNet* in comparison to other baselines for treatment effect estimation. The results of the experiment are presented in Table 1. Notably, *PoNet* outperforms the state-of-the-art methods in treatment effect estimation, as it effectively addresses the issue of post-treatment bias that is often neglected by other approaches.

**Verifying the effectiveness of inference policy**. Based on the previous analysis, the inference policy is to condition on the mediation post-treatment variables but not on the collider post-treatment variables. To validate the effectiveness of this policy, we introduce two variants of the inference policy in our model: (1) *PoNet with $Z_c$*, which conditions on the collider post-treatment variables $Z_c$ while also conditioning on the mediation post-treatment variables $Z_m$; (2) *PoNet w/o $Z_m$*, which neither conditions on the mediation post-treatment variables $Z_m$ nor on the collider post-treatment variables $Z_c$. We compare the performance of these two policy variants with that of the original inference policy, and the experimental results are illustrated in Figure 3. Please note that the standard deviation line has been scaled down to enhance the clarity of the results. The findings indicate that the two variants of the inference policy do not perform as well as the original policy. The measurement in terms of $\sqrt{\epsilon_{PEHE}}$ provides evidence of the existence of post-treatment bias, and it

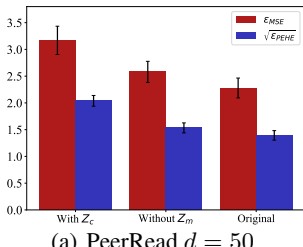
(a) PeerRead $d = 50$

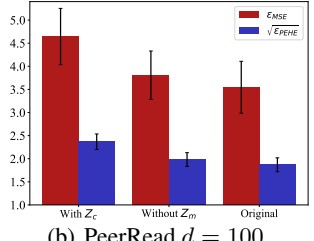
(b) PeerRead $d = 100$

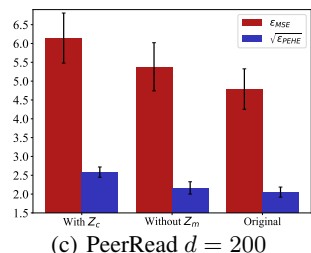
(c) PeerRead $d = 200$

Figure 3: Performance comparison between original inference policy and that w/o $Z_m$ or with $Z_c$.

further demonstrates that the proposed model with the original inference policy effectively captures the post-treatment variables and mitigates the post-treatment bias. Additionally, the measurement in terms of $\epsilon_{MSE}$ reveals that the decomposition of post-treatment variables contributes to the prediction of outcomes. It is intuitive that incorporating $Z_c$ into the outcome prediction introduces noise, while excluding $Z_m$ from the input results in the loss of valuable information, both of which can compromise the accuracy of the predictive model.

**Ablation Study**. We further investigate the impact of different components of the proposed model *PoNet* on the treatment effect estimation. Specifically, we conduct the ablation study

| Variants | $d = 50$ | $d = 100$ | $d = 200$ |
|---|---|---|---|
| *PoNet w/o CB* | $1.468 \pm 0.186$ | $1.919 \pm 0.464$ | $2.127 \pm 0.887$ |
| *PoNet w/o RM* | $1.521 \pm 0.186$ | $1.965 \pm 0.454$ | $2.157 \pm 0.910$ |
| *PoNet w/o MI* | $1.445 \pm 0.202$ | $1.939 \pm 0.514$ | $2.185 \pm 0.900$ |
| *PoNet* | $\mathbf{1.393 \pm 0.178}$ | $\mathbf{1.869 \pm 0.502}$ | $\mathbf{2.053 \pm 0.829}$ |

Table 2: Ablation Study on PeerRead in terms of $\sqrt{\epsilon_{PEHE}}$.

by deriving the following variants of the proposed model *PoNet*: (1) *PoNet w/o Confounder Balancing*, denoted by *PoNet w/o CB*; (2) *PoNet w/o Reconstruction Module*, denoted by *PoNet w/o RM*; (3) *PoNet w/o Mutual Information Regularizer*, denoted by *PoNet w/o MI*. We compare the performance of the three variants with the original model *PoNet*, the comparison results between the three variants and the original model are presented in Table 2. We can see that the original *PoNet* outperforms the other three variants, due to the following reasons: (1) *PoNet w/o CB* fails to adequately adjust for confounders, leading to confounding bias; (2) *PoNet w/o RM* is unable to effectively model the underlying factors, particularly the collider post-treatment variables that do not contribute to the outcome. Consequently, this can introduce potential post-treatment bias; (3) *PoNet w/o MI* is incapable of accurately separating the three underlying factors from each other, resulting in the potential generation of post-treatment and confounding bias.

## 4.4 REAL-WORLD DATA

MIMIC-III (Johnson et al., 2016) is a publicly available dataset of de-identified health-related data for over 40,000 patients who were admitted to the intensive care units of the Beth Israel Deaconess Medical Center between 2001 and 2012. The dataset includes data on patient demographics, vital signs, laboratory test results, medications, diagnoses, procedures, and imaging reports. Here we follow (Melnychuk et al., 2022) and use 25 vital signs and 3 static features as the covariates, whether using vasopressor as the treatment, the blood pressure as the outcome. Given that treatment can have an influence on numerous vital signs, it is essential to take into account the effects of post-treatment variables when estimating causal effects in real-world scenarios, particularly within healthcare data.

In order to showcase the experimental results, we randomly sampled data from two distinct time steps, denoted as $t_1$ and $t_2$, with each time step consisting of 6133 samples. Given that true counterfactuals are no longer accessible in real-world data, we evaluate the performance of predicting factual outcomes. The results are presented in Table 3. Our proposed model surpasses all state-of-the-art baselines, providing evidence of the superiority of our approach. In addition to excelling in causal effect estimation, our model also demonstrates strong performance in outcome prediction tasks. This can be attributed to our precise segmentation of covariates into distinct factors, thereby eliminating irrelevant variables from consideration. TEDVAE stands out among the baselines, showcasing commendable performance. This can be attributed to its variable decomposition approach, modeling irrelevant variables subsequently discarding them. However, an important limitation is its disregard for modeling post-treatment variables, which reduces its accuracy in prediction.

More importantly, we also want to verify if the proposed model *PoNet* can distinguish the three different underlying factors (e.g., confounders $C$, mediation post-treatment variables $Z_m$ and collider post-treatment variables $Z_c$). We employ t-SNE to reduce the dimensionality of the representations

Table 3: Performance of factual outcome prediction on real-world data MIMIC-III, lower is better.

| $\epsilon_{MSE}$ | $t_1$ | | $t_2$ | |
|---|---|---|---|---|
| | In-Sample | Out-of-Sample | In-Sample | Out-of-Sample |
| OLS1 | $0.351 \pm 0.004$ | $0.378 \pm 0.005$ | $0.410 \pm 0.004$ | $0.431 \pm 0.006$ |
| OLS2 | $0.330 \pm 0.004$ | $0.343 \pm 0.005$ | $0.393 \pm 0.005$ | $0.394 \pm 0.005$ |
| BART | $0.383 \pm 0.007$ | $0.335 \pm 0.023$ | $0.370 \pm 0.006$ | $0.368 \pm 0.019$ |
| Causal Forest | $0.361 \pm 0.012$ | $0.374 \pm 0.039$ | $0.404 \pm 0.013$ | $0.435 \pm 0.043$ |
| CEVAE | $0.315 \pm 0.003$ | $0.328 \pm 0.004$ | $0.357 \pm 0.003$ | $0.359 \pm 0.005$ |
| GANITE | $0.335 \pm 0.004$ | $0.327 \pm 0.003$ | $0.363 \pm 0.004$ | $0.361 \pm 0.005$ |
| TEDVAE | $0.284 \pm 0.003$ | $0.293 \pm 0.004$ | $0.304 \pm 0.004$ | $0.331 \pm 0.003$ |
| Tarnet | $0.302 \pm 0.004$ | $0.318 \pm 0.003$ | $0.340 \pm 0.004$ | $0.360 \pm 0.005$ |
| CFR | $0.301 \pm 0.003$ | $0.308 \pm 0.004$ | $0.339 \pm 0.003$ | $0.361 \pm 0.004$ |
| **PoNet** | $\mathbf{0.281 \pm 0.004}$ | $\mathbf{0.283 \pm 0.004}$ | $\mathbf{0.279 \pm 0.003}$ | $\mathbf{0.320 \pm 0.004}$ |

of the three underlying factors computed by the *PoNet model*. The representations are reduced to 2 dimensions and plotted using kernel density estimate to visualize the distribution of the three factors in the low-dimensional space as shown in Figure 4. The result clearly indicates that the inferred representations of the three factors from the proposed model exhibit significantly different distributions. This observation provides strong evidence for *Ponet*'s ability to effectively distinguish between the three underlying factors, even from the real-world cases

## 5    RELATED WORKS

Previous causal inference works mainly adjust confounders to control the confounding bias. Reweighting methods (Liu et al., 2020; Rosenbaum, 1987; Rosenbaum & Rubin, 1983; Wu et al., 2022) alter the instance weighting, such as by Inverse Propensity Weighting (IPW) (Glynn & Quinn, 2010), to create a more balanced comparison group. Stratification methods (Frangakis & Rubin, 2002; Hullsiek & Louis, 2002) divide the population into subgroups with similar covariate distributions to infer causal effects within each subgroup. Tree and forest-based methods like BART (Hill, 2011), Causal Forest (Wager & Athey, 2018), and Recursive partitioning (Athey & Imbens, 2016) estimate treatment effects for different subgroups of the population by building decision trees or random forests.

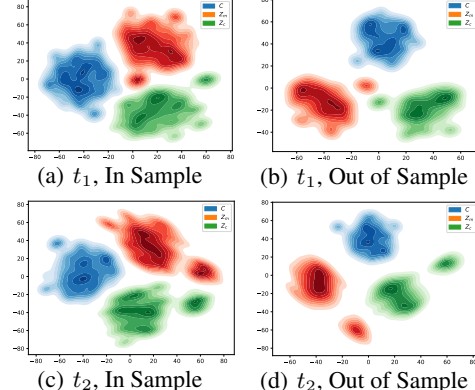

(a) $t_1$, In Sample    (b) $t_1$, Out of Sample

(c) $t_2$, In Sample    (d) $t_2$, Out of Sample

Figure 4: Visualization of distributions of inferred representations for confounders $C$, mediation post-treatment variables $Z_m$ and collider post-treatment variables $Z_c$.

Representation-based learning methods like CFR (Shalit et al., 2017), SITE (Yao et al., 2018), TAR-Net (Johansson et al., 2016), GANITE (Yoon et al., 2018), CEVAE (Louizos et al., 2017), TEDVAE (Zhang et al., 2021) map observed covariates to latent space, reducing the distribution discrepancy between treated and control groups, and have shown to be superior in estimating causal effects. The above methods assume that all variables are pre-treatment. However, post-treatment variables can introduce bias, as discussed in (Montgomery et al., 2018). Various studies (Coppock, 2019; Homola et al., 2020; King, 2010) outline ways to avoid post-treatment bias, but they mainly focus on experimental studies. A recent causal model (Li et al., 2022) addresses the inference of mediation post-treatment variables but does not consider collider post-treatment variables, thus potential collider post-treatment bias could be introduced.

## 6    CONCLUSION

In this study, we examine the sources and mechanisms of post-treatment bias and introduce a novel deep learning-based approach for decomposing variables and inferring post-treatment variables from observed covariates, utilizing a newly proposed causal graph specifically designed for post-treatment analysis. We also develop various components to infer the representations of confounders and post-treatment variables, thereby eliminating both confounding bias and post-treatment bias. Through extensive experiments on synthetic, semi-synthetic, and real-world datasets, we demonstrate the superior performance of our model compared to other state-of-the-art models in estimating heterogeneous treatment effects.

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
