# APPENDIX

## A ASSUMPTIONS

In causal inference, there are several basic assumptions that underlie the methods and techniques used to estimate causal effects. These assumptions provide a framework for reasoning about cause-and-effect relationships. Here we briefly introduce the assumptions used in this work:

**Assumption 1.** *Stable Unit Treatment Value Assumption. SUTVA states that the potential outcome of a unit (e.g. a person or an object) is not affected by the treatment assignment of any other unit. In other words, the assumption requires that the treatment assignment of one unit does not affect the potential outcome of any other unit.*

**Assumption 2.** *Unconfoundedness Assumption. This assumption states that, conditional on observed covariates, the treatment assignment is independent of the potential outcomes, i.e., $Y(T = 1), Y(T = 0) \perp T||X$. This means that there are no unobserved confounders that affect both the treatment assignment and the outcome.*

**Assumption 3.** *Positivity Assumption. This assumption states that there is a positive probability of receiving each treatment level, given the observed covariates, i.e., $0 < P(T = 1|X) < 1$. This means that there are no sub-populations for whom the treatment is impossible or infeasible and that the sample includes enough representation from each treatment level to estimate the treatment effect accurately.*

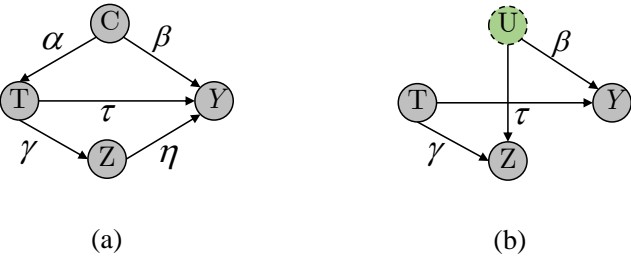

(a)                    (b)

Figure 5: The structural causal models for (a) Mediation post-treatment variables and (b) Collider post-treatment variables.

## B DERIVATION OF THE STRUCTURAL CAUSAL MODEL

In the preliminaries, we assume two types of the linear structural causal model (i.e., SCM) to illustrate two types of different post-treatment biases, here we present the detailed derivation of the average treatment effect estimation from the observational data for the two models:

(1) For the case of mediation post-treatment variables as presented in Figure 5. (a), the causal model can be formulated as follows:

$$Y = \tau T + \beta C + \eta Z = (\tau + \eta\gamma)T + \beta C, \tag{8}$$

we can see that the true total treatment effect of $T$ on $Y$ is $\tau + \eta\gamma$. Then the estimated average treatment effect from observational data can be derived as follows:

$$
\begin{aligned}
\Delta_a &= E(Y|T = 1) - E(Y|T = 0) \\
&= E(\tau T + \beta C + \eta Z|T = 1) - E(\tau T + \beta C + \eta Z|T = 0) \\
&= E(\tau T|T = 1) - E(\tau T|T = 0) + E(\beta C|T = 1) - E(\beta C|T = 0) + E(\eta Z|T = 1) - E(\eta Z|T = 0) \\
&= \tau(E(T|T = 1) - E(T|T = 0)) + \beta(E(C|T = 1) - E(C|T = 0)) + \eta(E(Z|T = 1) - E(Z|T = 0)) \\
&= \tau + \beta(E(C|T = 1) - E(C|T = 0)) + \eta(E(Z|T = 1) - E(Z|T = 0)).
\end{aligned}
\tag{9}
$$

(2) For the case of collider post-treatment variables as presented in Figure 5. (b), the causal model is formulated as follows:

$$Y = \tau T + \beta U, \tag{10}$$

the causal effect of $T$ on $Y$ is equal to $\tau$. Then the estimation of the average treatment effect from observational data for this model can be formulated as:

$$
\begin{aligned}
\Delta_b &= E(Y|T=1) - E(Y|T=0) \\
&= E(\tau T + \beta U|T=1) - E(\tau T + \beta U|T=0) \\
&= E(\tau T|T=1) - E(\tau T|T=0) + E(\beta U|T=1) - E(\beta U|T=0) \\
&= \tau(E(T|T=1) - E(T|T=0)) + \beta(E(U|T=1) - E(U|T=0)) \\
&= \tau + \beta(E(U|T=1) - E(U|T=0)).
\end{aligned}
\tag{11}
$$

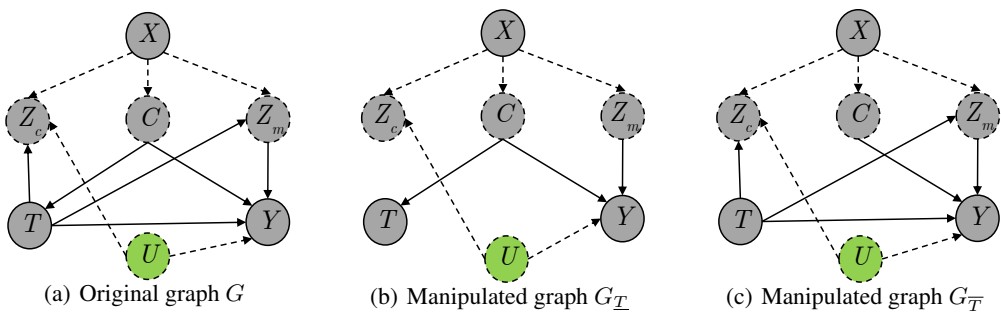

(a) Original graph $G$     (b) Manipulated graph $G_{\underline{T}}$     (c) Manipulated graph $G_{\overline{T}}$

Figure 6: The original causal graph $G$ and the corresponding manipulated causal graph.

## C   PROOF OF THEOREM 1

### C.1   RULES OF DO-CALCULUS

Here we adopt the causal graphical model to prove Theorem 1. Before we present the formal proof of Theorem 1, we need to introduce some basic concepts about the three rules of the do-calculusPearl (2012) in a causal graphical model.

*Given a causal graphical model, denoted by $G$ where $P$ denotes the associated distribution, let $Z, T, Y, W$ denote the arbitrary disjoint sets of variables in the graph $G$. Let $G_{\underline{T}}$ denote the graph with all edges out of $T$ removed, $G_{\overline{T}}$ denote the graph with all edges into $T$ removed, then the following three rules hold:*

- **Rule 1**: $P(Y|do(T), Z, W) = P(Y|do(T), Z)$ if $Y \perp W|(Z, T)$ in $G_{\overline{T}}$;

- **Rule 2**: $P(Y|do(T), Z) = P(Y|T, Z)$ if $Y \perp T|Z$ in $G_{\underline{T}}$;

- **Rule 3**: $P(Y|do(T), Z) = P(Y|Z)$ if $Y \perp T|Z$ in $G_{\overline{T}}$ and $Z$ is not a decedent of $T$.

### C.2   DERIVATION OF THE PROOF OF THEOREM 1

As shown in Figure 6, we present the original causal graphical model $G$ investigated in this work, $G_{\overline{T}}$ after removing all the edges into treatment $T$ and $G_{\underline{T}}$ after removing all the edges out of treatment $T$. The detailed derivation of the proof is as follows:

**Theorem 1:** *(Identifiability of Causal Effect under Post-treatment Variables and Confounders) If we can recover $p(Z_c|T, X)$, $p(z_m|T, X)$ and $p(C|X)$ from the observational data, then we can recover the causal effect of $T$ on $Y$ under post-treatment variables and confounders.*

*Proof.*

$$
\begin{aligned}
P(Y_u|do(T), X) &= P(Y|do(T), X, U) \\
&\overset{(i)}{=} \int_{Z_m} P(Y|do(T), X, U, Z_m) P(Z_m|do(T), X, U) \\
&\overset{(ii)}{=} \iint_{Z_m, C} P(Y|do(T), X, U, Z_m, C) P(C|do(T), X, U, Z_m) P(Z_m|do(T), X, U) \\
&\overset{(iii)}{=} \iiint_{Z_m, C, Z_c} P(Y|do(T), X, U, Z_m, C, Z_c) P(Z_c|do(T), X, U, Z_m, C) P(C|do(T), X, U, Z_m) P(Z_m|do(T), X, U) \\
&\overset{(iv)}{=} \iiint_{Z_m, C, Z_c} P(Y|do(T), U, Z_m, C) P(Z_c|do(T), X, U) P(C|do(T), X) P(Z_m|do(T), X, U) \\
&\overset{(v)}{=} \iiint_{Z_m, C, Z_c} P(Y|T, U, Z_m, C) P(Z_c|T, X, U) P(C|do(T), X) P(Z_m|T, X) \\
&\overset{(vi)}{=} \iiint_{Z_m, C, Z_c} P(Y|T, U, Z_m, C) P(Z_c|T, X, U) P(C|X) P(Z_m|T, X) \\
&\overset{(vii)}{=} \iint_{Z_m, C} P(Y|T, U, Z_m, C) P(C|X) P(Z_m|T, X) \\
&\overset{(vii)}{=} \iint_{Z_m, C} P(Y_u|T, Z_m, C) P(C|X) P(Z_m|T, X)
\end{aligned}
\tag{12}
$$

where $Y_u$ is the observed outcome with $U = u$; the equation $(i)$, $(ii)$ and $(iii)$ are the straightforward expectation over $P(Z_m|do(T), X)$, $P(C|do(T), X, Z_m)$ and $P(Z_c|do(T), X, Z_m, C)$, respectively; equation $(iv)$ is derived by the **Rule 1** given the following independence conditions $\{Y \perp X, Z_c|(T, U, Z_m, C)\}$, $\{Z_c \perp Z_m, C|(T, X, U)\}$, $\{C \perp U, Z_m|(T, X)\}$ and $\{Z_m \perp U\}$ in the graph $G_{\overline{T}}$; equation $(v)$ is derived by the **Rule 2** given the following independence conditions $\{Y \perp T|U, Z_m, C\}$, $\{Z_c \perp T|X, U\}$ and $\{Z_m \perp T|X\}$ in the graph $G_{\underline{T}}$; equation $(vi)$ is derived by the **Rule 3** given the independence condition $\{C \perp T|X\}$ in the graph $G_{\overline{T}}$; equation $(vii)$ is derived by the integration over $Z_c$; Noting that the above independence condition can be obtained from the $d$-separation criterion Geiger et al. (1990). □

## C.3 DERIVATION OF THE PROOF OF THEOREM 2

**Theorem 2:** *The joint set of inferred factors for $C$, $Z_m$ is minimally sufficient for the optimal parameters $\theta$ which estimation of unbiased treatment effects needs.*

To simplify our proof, we collectively denote the underlying factors $\{C, Z_m\}$ as the statistic $S = S(X_1, ..., X_d)$ where $(x_1, x_2..., x_d)$ is the original covariates of the observed sample. Let $\theta$ represent the optimal parameter instantiation that the treatment effect can be estimated. To demonstrate that $S = (C, Z_m)$ is the minimally sufficient statistic to infer the unbiased treatment effect, it must be proven that the statistic $S$ is minimally sufficient for $\theta$. We first introduce two lemmas about the minimally sufficient statistic Silvey (2017):

**Lemma 1.** *(Sufficient statistic). A statistic $S = S(X_1, ..., X_d)$ is sufficient for $\theta$ if the distribution of the sample given $S$, i.e., $\mathbb{P}(X_1, ..., X_d|S)$, does not depend on $\theta$.*

**Lemma 2.** *(Minimal sufficient statistic). A sufficient statistic $S$ for $\theta$ is minimal sufficient if there exist a measurable function $\phi$ such that $S = \phi(\bar{S})$ for any other sufficient statistic $\bar{S}$*

**Lemma 3.** *(Independence equivalence). Given any two sample realizations $(x_1, ..., x_d)$ and $(x'_1, ..., x'_d)$, the following equivalence holds:*

$$
\frac{\mathcal{L}(x_1, ..., x_d; \theta)}{\mathcal{L}(x'_1, ..., x'_d; \theta)} \quad is\, independent \quad of \quad \theta \Leftrightarrow S(x_1, ..., x_d) = S(x'_1, ..., x'_d),
$$

*Proof.* First, we prove that the statistic $S$ is sufficient. For any sample $(x'_1, ..., x'_d)$, we have:

$$
\mathbb{P}(X_1 = x'_1, ..., X_d = x'_d) = \begin{cases} 0, & \text{if } S(x'_1, ..., x'_d) \neq s \\ \frac{\mathbb{P}(X_1 = x'_1, ..., X_d = x'_d; \theta)}{\mathbb{P}(S = s; \theta)}, & \text{if } S(x'_1, ..., x'_d) = s, \end{cases}
$$

$\square$

For the samples such that $S(x'_1, ..., x'_d) \neq s$, the distribution $\mathbb{P}(X_1, .., X_d | S)$ is clearly independent of $\theta$. Then for the samples such that $S(x'_1, ..., x'_d) = s$, we have the following derivation:

$$
\begin{aligned}
\mathbb{P}(X_1 = x'_1, ..., X_d = x'_d | S = s) &= \frac{\mathbb{P}(X_1 = x'_1, ..., X_d = x'_d; \theta)}{\mathbb{P}(S = s; \theta)} \\
&= \frac{\mathbb{P}(X_1 = x'_1, ..., X_d = x'_d; \theta)}{\sum_{(x_1, ..., x_d) \in A_s} \mathbb{P}(X_1 = x1, ..., X_d = x_d; \theta)} \\
&= \frac{\mathcal{L}(x'_1, ..., x'_d; \theta)}{\sum_{(x_1, ..., x_d) \in A_s} \mathcal{L}(x_1, ..., x_d; \theta)} \\
&= \frac{1}{\sum_{(x_1, ..., x_d) \in A_s} \frac{\mathcal{L}(x_1, ..., x_d; \theta)}{\mathcal{L}(x'_1, ..., x'_d; \theta)}}
\end{aligned}
\tag{13}
$$

where $A_s = \{(x_1, ..., x_d) \in (R)^d : S(x_1, ..., x_d) = s\}$. In this case, the samples $(x_1, ..., x_d)$ and $(x'_1, ..., x'_d)$ share the same value $s$ of the statistic $S$, thus the ratio of likelihoods $\frac{\mathcal{L}(x_1, ..., x_d; \theta)}{\mathcal{L}(x'_1, ..., x'_d; \theta)}$ in the denominator does not depend on based on the Lemma 3, then the distribution $\mathbb{P}(X_1 = x_1, ..., X_d = x_d | S = s)$ is also independent of $\theta$, thus the statistic $S$ is sufficient for $\theta$ according to Lemma 1, which means the inferred factors $C, Z_m$ is sufficient for the optimal treatment effect estimation.

Next, we prove the minimal sufficiency of the statistic $S$. Let $\bar{S}$ be another sufficient statistic for $\theta$, based on Lemma 2, we have to prove that there exists a function $\phi$ that $S = \phi(\bar{S})$. Similarly, let $(x_1, ..., x_d)$ and $(x'_1, ..., x'_d)$ be two samples that share the same value for the sufficient statistic $\bar{S}$, i.e., $\bar{S}(x_1, ..., x_d) = \bar{S}(x'_1, ..., x'_d) = \bar{s}$, thus the following two probabilities of such samples given $\bar{S} = \bar{s}$ are both independent of $\theta$:

$$
\begin{aligned}
\mathbb{P}(X_1 = x_1, ..., X_d = x_d | \bar{S} = \bar{s}) &= \frac{\mathbb{P}(X_1 = x_1, ..., X_d = x_d; \theta)}{\mathbb{P}(\bar{S} = \bar{s}; \theta)}, \\
\mathbb{P}(X_1 = x'_1, ..., X_d = x'_d | \bar{S} = \bar{s}) &= \frac{\mathbb{P}(X_1 = x'_1, ..., X_d = x'_d; \theta)}{\mathbb{P}(\bar{S} = \bar{s}; \theta)},
\end{aligned}
\tag{14}
$$

thus the likelihood ratio

$$
\frac{\mathbb{P}(X_1 = x_1, ..., X_d = x_d; \theta)}{\mathbb{P}(X_1 = x'_1, ..., X_d = x'_d; \theta)} = \frac{\mathcal{L}(x_1, ..., x_d; \theta)}{\mathcal{L}(x'_1, ..., x'_d; \theta)}
$$

is also independent of $\theta$. According to Lemma 3, we have

$$
S(x_1, ..., x_d) = S(x'_1, ..., x'_d)
$$

, then we can see that the two samples share the same value of both $S$ and $\bar{S}$, that is, for each value $\bar{s}$ of $\bar{S}$, there is a unique value $\phi(\bar{s})$, and thus $S = \phi(\bar{S})$. Based on Lemma 2, the statistic $S$ is minimally sufficient for $\theta$.

## D DETAILED STRUCTURE OF *PoNet*

In this work, we propose a novel deep learning-based model *PoNet*, which aims to decompose the confounders and two types of post-treatment variables from the observational data for eliminating post-treatment bias and confounding bias for treatment effect estimation. The structure overview of *PoNet* is shown in Figure 7.

**Learning function for inferring the three underlying factors**. Here details the formulation for learning the representations of post-treatment variables, collider post-treatment variables, and

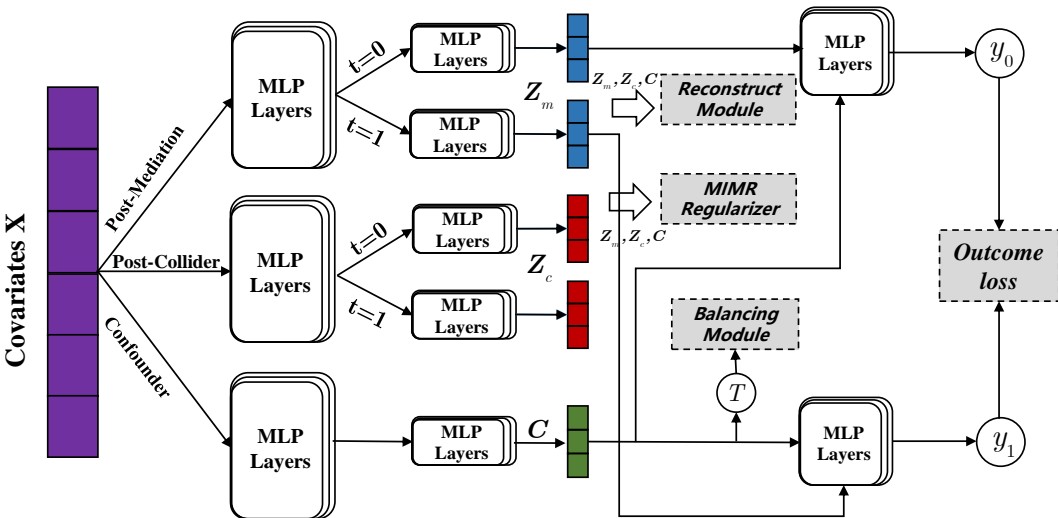

Figure 7: The structure overview of the proposed model *PoNet*.

confounders. First, the function $f_{me}(\boldsymbol{x}, t)$ for learning the representations of the mediation post-treatment variables, which is parameterized by stacking $L$ fully connected layers, can be formulated as follows:

$$\boldsymbol{z}_m = f_{me}(\boldsymbol{x}, t) = \begin{cases} \varphi(\boldsymbol{W}_L^0 ... \varphi(\boldsymbol{W}_1^0 \boldsymbol{x} + c_1^0) + c_L^0) & \text{if} \quad t = 0 \\ \varphi(\boldsymbol{W}_L^1 ... \varphi(\boldsymbol{W}_1^1 \boldsymbol{x} + c_1^1) + c_L^1) & \text{if} \quad t = 1, \end{cases} \quad (15)$$

where $\boldsymbol{x}$ is the observed covariates and $t$ is the assigned treatment, $\varphi(\cdot)$ denotes the activation function, $\boldsymbol{W}_K^t$ and $c_K^t$ are the learning weight and bias term for the $K$-th hidden layer. Note that we can formulate the function $f_{co}(\boldsymbol{x}, t)$ for learning the representations $\boldsymbol{z}_c$ of collider post-treatment variable in the similar formulation.

Similarly, the function $f_c(\boldsymbol{x})$ for learning the representations of confounders, which is parameterized by stacking $L$ fully connected layers, can be formulated as follows:

$$\boldsymbol{c} = \varphi(\boldsymbol{W}_L^c ... \varphi(\boldsymbol{W}_1^c \boldsymbol{x} + c_1^c) + c_L^c), \quad (16)$$

where $\boldsymbol{c}$ denotes the representation of confounders.

**Outcome prediction network**. Incorporating the inferred representations of the mediation post-treatment variables and confounders, the outcome prediction network can be parameterized by fully connected layers, which can be formulated as follows:

$$f_y(\boldsymbol{c}^i, \boldsymbol{z}_m^i, t_i) = \begin{cases} \hat{y}_i^{t_i=0} = f_0(\boldsymbol{c}^i, \boldsymbol{z}_m^i) & \text{if } t_i = 0 \\ \hat{y}_i^{t_i=1} = f_1(\boldsymbol{c}^i, \boldsymbol{z}_m^i) & \text{if } t_i = 1, \end{cases} \quad (17)$$

where $f_0(\cdot)$ and $f_1(\cdot)$ are two output functions that are parameterized by MLPs for treatment $t_i = 0$ and $t_i = 1$, respectively.

**Balancing Module**. Here we present the details about the representation balancing module. Here we utilize Wasserstein disance, which is a variant of the optimal transport distance, to measure the minimum amount of work required to transform one distribution into another one. Let $P(\boldsymbol{c}) = Pr(\boldsymbol{c}|t = 1)$ and $Q(\boldsymbol{c}) = Pr(\boldsymbol{c}|t = 0)$ denote the empirical distributions of representation of confounders for treated and control units, respectively, $\mathcal{S}_{\mathcal{Z}}(P, Q)$ represent the Wasserstein distance defined in the 1-Lipschitz functional space $\mathcal{Z}$, the balancing term can be formulated as follows:

$$\mathcal{L}_{wass} = \mathcal{S}_{\mathcal{Z}(P,Q)} = \inf_{k \in \mathcal{K}} \int_{\boldsymbol{c} \in \{\boldsymbol{c}_i\}_{i:t_i=1}} ||k(\boldsymbol{c}) - \boldsymbol{c}|| P(\boldsymbol{c}) d\boldsymbol{c} \quad (18)$$
$$s.t. \quad Q(k(\boldsymbol{c})) = P(\boldsymbol{c}),$$

where $\mathcal{K} = \{k|k : \mathbb{R}^m \to \mathbb{R}^m\}$ denotes the set of push-forward functions that can transform the distribution of the treated units' representations to that of the control units' representations. And we use the approximation algorithm proposed by (Cuturi & Doucet, 2014) to compute the Wasserstein-1 distance $\mathcal{S}_{\mathcal{Z}(P,Q)}$.

# E  EXPERIMENTS

## E.1  EXPERIMENT SETTINGS

For each dataset, we run the experiments for 10 times and report the average performance in terms of specified metrics. For each evaluation, we spilt the dataset into the training set (80%) and the test set (20%). Regarding the hyperparameters of the proposed model, we adopt the grid search strategy to find the optimal parameter combination. Specifically, we set the learning rate as 0.01, the weight $\alpha$ of the MIMR regularizer for separating the underlying factors to range in 0.0001, 0.001, 0.01, 0.1, 1}, the weight $\beta$ of the covariate reconstruction module to range in $\{0.0001, 0.001, 0.01, 0.1, 1\}$, the weight $\gamma$ of balancing confounders by Wasserstein distance to range in $\{0.0001, 0.001, 0.01, 0.1, 1\}$, the weight $\eta$ for controlling the over-fitting as 0.0001, the number of hidden layers of each representation network and prediction network to range in $\{1, 2, 3, 4\}$, the dimension $m$ of each hidden layer to range in $\{50, 100, 150, 200\}$.

We use the Stochastic Gradient Descent (SGD) or Adam as the optimizer to train the models. All codes are implemented in Python, the deep learning framework we adopt in this work is Pytorch. And we use Intel(R) Xeon(R) Gold 5120 CPU @ 2.20GHz and 512G, NVIDIA TITAN RTX GPU @24 GB. The anonymous link of the source code of the proposed model *PoNet* is: `https://anonymous.4open.science/r/Ponet-37F2/`.

## E.2  SYNTHETIC DATA

### E.2.1  DATA GENERATION PROCESS

We generate the synthetic data according to the following process. The input of the data generation is the sample size $N$; the dimension $d_C$, $d_{Z_m}$, $d_{Z_c}$ for each underlying factor $\{C, Z_m, Z_c\}$, respectively; then the generation can be formulated as the following steps:

- **Step 1**: Draw $N$ samples with the size $d_C$ from the Gaussian distribution $\mathcal{N}(\mu_C, \Sigma_C)$ where $\mu_C$ and $\Sigma_C$ means and covariance matrix, to form the confounder factor $\boldsymbol{x}_C \in \mathbb{R}^{N \times d_C}$.

- **Step 2**: Draw $N$ samples with the size $d_{Z_m}$ from the Gaussian distribution $\mathcal{N}(\mu_{Z_{m1}}, \Sigma_{Z_{m1}})$ to form the factors $\boldsymbol{x}_{Z_{m1}}$ for treatment $T = 1$, and draw $N$ samples with the size $d_{Z_m}$ from the Gaussian distribution $\mathcal{N}(\mu_{Z_{m0}}, \Sigma_{Z_{m0}})$ to form the factors $\boldsymbol{x}_{Z_{m0}}$ for treatment $T = 0$;

- **Step 3**: Draw $N$ samples with the size $d_{Z_c}$ from the Gaussian distribution $\mathcal{N}(\mu_{Z_{c1}}, \Sigma_{Z_{c1}})$ to form the factors $\boldsymbol{x}_{Z_{c1}}$ for treatment $T = 1$, and draw $N$ samples with the size $d_{Z_c}$ from the Gaussian distribution $\mathcal{N}(\mu_{Z_{c0}}, \Sigma_{Z_{c0}})$ to form the factors $\boldsymbol{x}_{Z_{c0}}$ for treatment $T = 0$;

- **Step 4**: Sample the coefficients $\boldsymbol{w}_t \in \mathbb{R}^{d_C}$ from the distribution $\mathcal{N}(0, 1)$; then formulate the treatment assignment policy as $\Delta(T = 1|\boldsymbol{x}_C) = \frac{1}{1 + exp(-\boldsymbol{w}_t \boldsymbol{x}_C)}$, then we can obtain the treatment assignments $\{t_1, t_2, ..., t_N\}$ for $N$ samples based on the Bernoulli distribution with parameter $\Delta(T = 1|\boldsymbol{x}_C)$.

- **Step 5**: According to the obtained treatment assignment, form the observed mediation post-treatment variables $\boldsymbol{x}_{Z_m}$ from $\boldsymbol{x}_{Z_{m1}}$ or $\boldsymbol{x}_{Z_{m0}}$, the observed collider post-treatment variables $\boldsymbol{x}_{Z_c}$ from $\boldsymbol{x}_{Z_{c1}}$ or $\boldsymbol{x}_{Z_{c0}}$ for each sample.

- **Step 6**: Concatenate the confounders $\boldsymbol{x}_C$, meditation post-treatment variables $\boldsymbol{x}_{Z_m}$ and collider post-treatment variables $\boldsymbol{x}_{Z_c}$ to be the observed covariates $\boldsymbol{x} = \{\boldsymbol{x}_C, \boldsymbol{x}_{Z_m}, \boldsymbol{x}_{Z_c}\}$ for each sample.

- **Step 7**: Sample the two coefficient tuples $\boldsymbol{w}_0 \in \mathbb{R}^{d_C + d_{Z_m}}$ and $\boldsymbol{w}_1 \in \mathbb{R}^{d_C + d_{Z_m}}$ from the distribution $\mathcal{N}(0, 1)$, then formulate the control outcome as $y^0 = (\boldsymbol{x}_{CZ_{m0}} \circ \boldsymbol{x}_{CZ_{m0}})^T \cdot \boldsymbol{w}_0/(d_C + d_{Z_m})$ and treated outcome as $y^1 = (\boldsymbol{x}_{CZ_{m1}} \circ \boldsymbol{x}_{CZ_{m1}} \circ \boldsymbol{x}_{CZ_{m1}})^T \cdot \boldsymbol{w}_1/(d_C + d_{Z_m})$, where $\boldsymbol{x}_{CZ_{mi}}(i = 0, 1)$ is the concatenation of $\{\boldsymbol{x}_C, \boldsymbol{x}_{Z_{mi}}\}$ and $\circ$ means the Hadamard product.

### E.2.2  ADDITIONAL EXPERIMENTS

**Treatment effect and prediction performance on synthetic data in terms of $\sqrt{\epsilon_{PEHE}}$ and $\epsilon_{MSE}$.** Here we report the performance of the proposed model *PoNet* on treatment effect estimation

Table 4: Performance of different models on treatment effect estimation and prediction with size $N = 8000$ on synthetic data, lower is better.

| $N = 8000$ | $d = 8$ | | $d = 16$ | | $d = 24$ | |
|---|---|---|---|---|---|---|
| | $\sqrt{\epsilon_{PEHE}}$ | $\epsilon_{MSE}$ | $\sqrt{\epsilon_{PEHE}}$ | $\epsilon_{MSE}$ | $\sqrt{\epsilon_{PEHE}}$ | $\epsilon_{MSE}$ |
| *OLS1* | 0.9682 | 0.3097 | 0.5189 | 0.1107 | 0.4265 | 0.0695 |
| *OLS2* | 0.6796 | 0.2235 | 0.4202 | 0.0874 | 0.3390 | 0.0515 |
| *BART* | 0.5826 | 0.0940 | 0.4820 | 0.0777 | 0.4169 | 0.0666 |
| *Causal Forest* | 0.4687 | 0.1660 | 0.4058 | 0.0839 | 0.3819 | 0.0859 |
| *CEVAE* | 0.7934 | 0.3262 | 0.4666 | 0.0906 | 0.4212 | 0.0763 |
| *GANITE* | 0.7821 | 0.1855 | 0.476 | 0.0923 | 0.3714 | 0.0592 |
| *TEDVAE* | 0.5574 | **0.0123** | 0.3862 | 0.0297 | 0.4843 | 0.0427 |
| *Tarnet* | 0.3335 | 0.0124 | 0.3499 | 0.0350 | 0.4140 | 0.0532 |
| *CFR* | 0.3251 | 0.0143 | 0.3527 | 0.0384 | 0.4078 | 0.0540 |
| ***PoNet*** | **0.2572** | 0.0143 | **0.2760** | **0.0151** | **0.3130** | **0.0275** |

Table 5: Performance of different models on treatment effect estimation and prediction with size $N = 15000$ on synthetic data, lower is better.

| $N = 15000$ | $d = 8$ | | $d = 16$ | | $d = 24$ | |
|---|---|---|---|---|---|---|
| | $\sqrt{\epsilon_{PEHE}}$ | $\epsilon_{MSE}$ | $\sqrt{\epsilon_{PEHE}}$ | $\epsilon_{MSE}$ | $\sqrt{\epsilon_{PEHE}}$ | $\epsilon_{MSE}$ |
| *OLS1* | 0.9684 | 0.3079 | 0.5100 | 0.1047 | 0.4354 | 0.0743 |
| *OLS2* | 0.6814 | 0.2262 | 0.4063 | 0.0803 | 0.3404 | 0.0538 |
| *BART* | 0.6466 | 0.1391 | 0.4773 | 0.0810 | 0.4144 | 0.0721 |
| *Causal Forest* | 0.4588 | 0.1492 | 0.3968 | 0.0750 | 0.3732 | 0.0785 |
| *CEVAE* | 0.6921 | 0.2688 | 0.4636 | 0.0957 | 0.4155 | 0.0746 |
| *GANITE* | 0.7587 | 0.1652 | 0.4733 | 0.0837 | 0.3953 | 0.0677 |
| *TEDVAE* | 0.6056 | 0.0158 | 0.3582 | 0.0260 | 0.4064 | 0.0286 |
| *Tarnet* | 0.2949 | 0.0077 | 0.2805 | 0.0119 | 0.3780 | 0.0379 |
| *CFR* | 0.2840 | 0.0096 | 0.2950 | 0.0167 | 0.3821 | 0.0387 |
| ***PoNet*** | **0.2262** | **0.0037** | **0.2386** | **0.0067** | **0.2533** | **0.0136** |

and prediction comparing to other baselines on the synthetic data. We set the dimension of each underlying factors as $d = \{8, 16, 24\}$ and generate $N = 8000$ and $15000$ samples. The experimental results on synthetic data are shown in Table 4 and 5. One can see that the performance of the proposed model *PoNet* is better than that of other models in general.

**Treatment effect Performance in terms of $\epsilon_{ATE}$.** Here we report the error on ATE $\epsilon_{ATE} = |\frac{1}{N}\sum_{i=1}^{N}(y_i^{t_i=1} - y_i^{t_i=0}) - \frac{1}{N}\sum_{i=1}^{N}(\hat{y}_i^{t_i=1} - \hat{y}_i^{t_i=0})|$. The performance on metric $\epsilon_{ATE}$ is shown in Table 6. As we can see that the proposed model *PoNet* outperforms the other baselines in most cases.

**How important the Reconstruction Module is**. Here we show that the proposed reconstruction module plays an important role in identifying and recovering collider post-treatment variables. In our hypothesized causal mechanism, the collider post-treatment variables have no effect on the outcome, thus there is lack of supervised information to guide the learning of representations of this factor. The experiment in the section 4.2 in main body for demonstrating the capability of identifying each underlying factor shows that the proposed model *PoNet* can identify the collider post-treatment variables well. But what if there is no reconstruction module? Here we remove the reconstruction module from the model and see what the result looks like for identifying the collider post-treatment variables. We adopt the similar experiment setting and design as in section 4.2, the radar plots before and after removing the reconstruction module are show in Figure 8. As we can see that the result shows that the reconstruction module is essential for identifying and recovering the underlying factor which is not contributed to the outcome.

Table 6: Performance comparison in terms of $\epsilon_{ATE}$ on the synthetic dataset, lower is better.

| Synthetic | N=8000 | | | N=15000 | | |
|---|---|---|---|---|---|---|
| | $d$=8 | $d$=16 | $d$=24 | $d$=8 | $d$=16 | $d$=24 |
| LR | 0.100 | 0.055 | 0.088 | 0.102 | 0.058 | 0.093 |
| OLS2 | 0.101 | 0.074 | 0.056 | 0.047 | 0.071 | 0.064 |
| BART | 0.102 | 0.0383 | 0.040 | 0.089 | 0.051 | **0.029** |
| Causal Forest | 0.077 | 0.069 | 0.050 | 0.043 | 0.049 | 0.048 |
| CEVAE | 0.088 | 0.178 | 0.229 | 0.103 | 0.168 | 0.221 |
| GANITE | 0.121 | 0.082 | 0.066 | 0.083 | 0.144 | 0.111 |
| TEDVAE | 0.102 | **0.040** | 0.062 | 0.053 | 0.073 | 0.047 |
| Tarnet | 0.072 | 0.056 | 0.043 | 0.076 | 0.039 | 0.073 |
| CFR | 0.067 | 0.042 | 0.037 | 0.061 | 0.055 | 0.063 |
| PoNet | **0.059** | 0.048 | **0.037** | **0.041** | **0.035** | 0.047 |

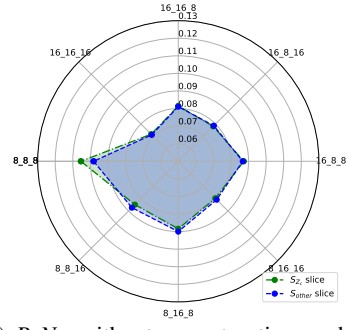

(a) *PoNet* without reconstruction module

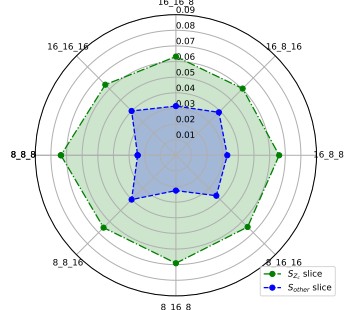

(b) *PoNet* with reconstruction module

Figure 8: The radar charts for identifying $Z_c$ before and after removing the reconstruction module.

### E.3 SEMI-SYNTHETIC DATA

#### E.3.1 DATA GENERATION PROCESS

The semi-synthetic dataset PeerRead used in this work consists of 7601 instances, associated with 1080 covariates. Each instance represents an author, the covariates of each instance are bag-of-word representations extracted from their papers' titles and abstracts.

We follow the semi-synthetic simulation proposed in Johansson et al. (2016). We assume the dimension of mediation post-treatment variables and collider post-treatment variables are $d_{Z_m}$ and $d_{Z_c}$, and denote the associated features as $\boldsymbol{x}$. First we train a Latent Dirichlet Allocation (LDA) topic model with 50 topics to map the covariates of each instance to the topic space, the topic distribution of each instance $x$ is denoted by $\boldsymbol{z}(x)$. Then we define two centroids, one is formulated as the average topic representations denoted by $\boldsymbol{z}_0$, another one is the topic distribution of a randomly sampled instance ,denoted by $\boldsymbol{z}_1$. The treatment assignment policy can be defined as follows:

$$\Delta(T|\boldsymbol{z}(x)) = \frac{e^{k\boldsymbol{z}_1^T\boldsymbol{z}(x)}}{e^{k\boldsymbol{z}_1^T\boldsymbol{z}(x)} + e^{k\boldsymbol{z}_0^T\boldsymbol{z}(x)}}, \tag{19}$$

where $k$ controls the magnitude of the confounding bias. The treatment assignment of each instance is generated by the Bernoulli distribution with the above parameter. Then we generate the post-treatment factors based on the treatment assignment policy:

(1) Draw $N = 7601$ with the size $d_{Z_m}$ from the Gaussian distribution $\mathcal{N}(\mu_{Z_{m0}}, \Sigma_{Z_{m0}})$ to form the mediation post-treatment factors $\boldsymbol{x}_{Z_{m0}}$ for treatment $T = 0$; draw $N = 7601$ with the size $d_{Z_m}$ from the Gaussian distribution $\mathcal{N}(\mu_{Z_{m1}}, \Sigma_{Z_{m1}})$ to form the mediation post-treatment factors $\boldsymbol{x}_{Z_{m1}}$ for treatment $T = 1$;

(2) Draw $N = 7601$ with the size $d_{Z_c}$ from the Gaussian distribution $\mathcal{N}(\mu_{Z_{c0}}, \Sigma_{Z_{c0}})$ to form the mediation post-treatment factors $\boldsymbol{x}_{Z_{c0}}$ for treatment $T = 0$; draw $N = 7601$ with the size $d_{Z_c}$

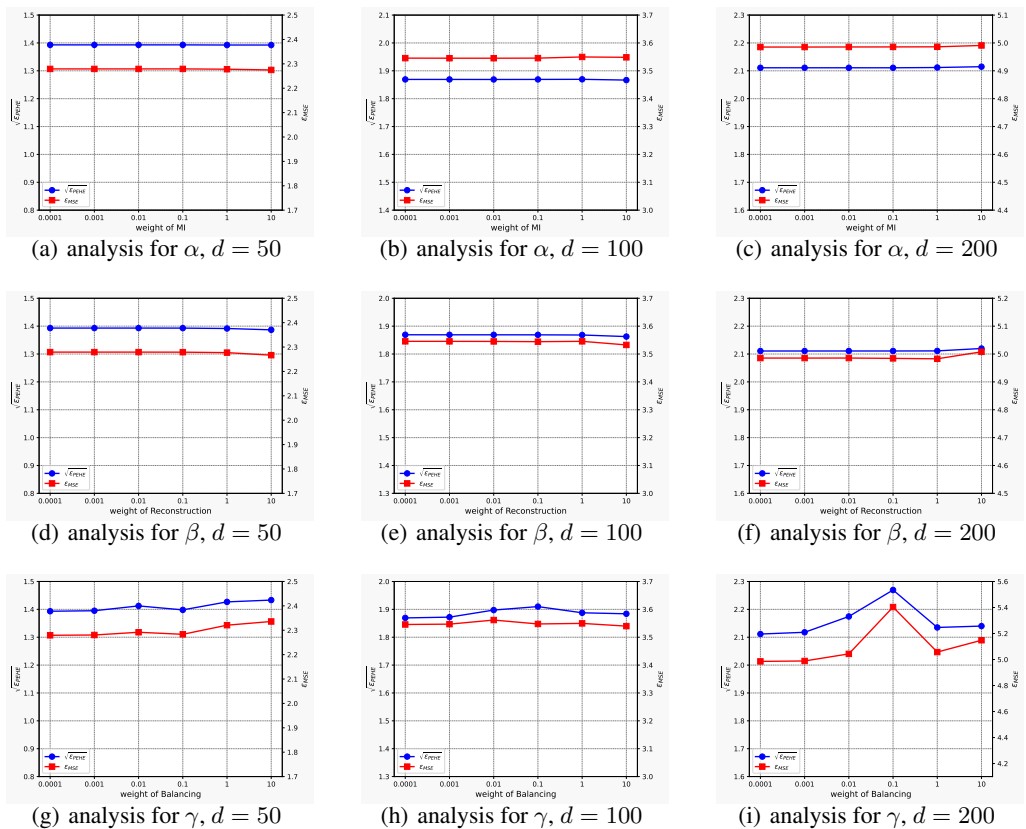

Figure 9: Hyper-parameter analysis to explore the impacts of different weights of mutual information regularizer, reconstruction module, and confounder balancing module.

from the Gaussian distribution $\mathcal{N}(\mu_{Z_{c1}}, \Sigma_{Z_{c1}})$ to form the mediation post-treatment factors $\boldsymbol{x}_{Z_{c1}}$ for treatment $T = 1$;

(3) According to the generated treatment assignments, simulate the observed mediation and collider post-treatment variables as $\boldsymbol{x}_{Z_m}$ and $\boldsymbol{x}_{Z_c}$ from the formed $\{\boldsymbol{x}_{Z_{m0}}, \boldsymbol{x}_{Z_{m1}}\}$ and $\{\boldsymbol{x}_{Z_{c0}}, \boldsymbol{x}_{Z_{c1}}\}$. Then combine the observed features $\boldsymbol{x}$, generated mediation post-treatment variables $\boldsymbol{x}_{Z_m}$ and collider post-treatment variables $\boldsymbol{x}_{Z_c}$ as the covariates.

(4) The simulation of the potential outcome can be formulated as follows:

$$
\begin{aligned}
y^0 &= C \cdot (k\boldsymbol{z}_0^T \boldsymbol{z}(x)) + \boldsymbol{w}_0^T \cdot (\boldsymbol{x}_{Z_{m0}} \circ \boldsymbol{x}_{Z_{m0}})/d_{Z_m} + \epsilon, \\
y^1 &= C \cdot (k\boldsymbol{z}_1^T \boldsymbol{z}(x) + k\boldsymbol{z}_0^T \boldsymbol{z}(x)) + \boldsymbol{w}_1^T \cdot (\boldsymbol{x}_{Z_{m1}} \circ \boldsymbol{x}_{Z_{m1}} \circ \boldsymbol{x}_{Z_{m1}})/d_{Z_m} + \epsilon,
\end{aligned}
\tag{20}
$$

where $C$ is the scaling factor, $\boldsymbol{w}_0$ and $\boldsymbol{w}_1$ are the coefficients draw from the distribution $\mathcal{N}(1, 1)$ and $\epsilon$ is the white noise draw from the distribution $\mathcal{N}(0, 1)$.

### E.3.2 ADDITIONAL EXPERIMENTS

**Performance in terms of** $\epsilon_{ATE}$. Here we report the error on ATE $\epsilon_{ATE}$. The performance on metric $\epsilon_{ATE}$ is shown in Table 7. As we can see that the proposed model *PoNet* outperforms the other baselines.

**Performance comparison when removing the post-treatment bias.**

**Hyper-parameter Study**. In this section, we perform a comprehensive hyper-parameter study to explore the impact of different parameter settings on the performance of our model. Hyper-parameters play a crucial role in machine learning algorithms as they control the behavior and flexibility of the

Table 7: Performance comparison in terms of $\epsilon_{ATE}$ on the semi-synthetic dataset, lower is better.

| Semi-synthetic PeerRead | d=50 | d=100 | d=200 |
|---|---|---|---|
| LR | $0.405 \pm 0.164$ | $0.317 \pm 0.343$ | $1.140 \pm 0.785$ |
| OLS2 | $0.147 \pm 0.067$ | $0.363 \pm 0.074$ | $0.723 \pm 0.108$ |
| BART | $0.209 \pm 0.089$ | $0.445 \pm 0.241$ | $1.253 \pm 0.613$ |
| Causal Forest | $0.172 \pm 0.066$ | $0.386 \pm 0.183$ | $0.938 \pm 0.390$ |
| CEVAE | $0.415 \pm 0.079$ | $0.544 \pm 0.125$ | $0.761 \pm 0.357$ |
| GANITE | $1.740 \pm 0.390$ | $1.748 \pm 0.243$ | $0.506 \pm 0.396$ |
| TEDVAE | $0.339 \pm 0.169$ | $0.399 \pm 0.314$ | $0.599 \pm 0.639$ |
| Tarnet | $1.018 \pm 0.210$ | $0.498 \pm 0.198$ | $0.757 \pm 0.246$ |
| CFR | $1.045 \pm 0.181$ | $0.544 \pm 0.220$ | $0.706 \pm 0.233$ |
| PoNet | $\mathbf{0.125 \pm 0.058}$ | $\mathbf{0.131 \pm 0.077}$ | $\mathbf{0.138 \pm 0.137}$ |

Table 8: Estimation performance comparison when removing post-treatment bias on semi-synthetic PeerRead.

| | $\sqrt{\epsilon_{PEHE}}$ | $\epsilon_{ATE}$ | $\epsilon_{MSE}$ |
|---|---|---|---|
| LR | $2.718 \pm 1.076$ | $0.206 \pm 0.089$ | $6.647 \pm 4.495$ |
| OLS2 | $2.279 \pm 0.852$ | $0.102 \pm 0.113$ | $6.403 \pm 4.352$ |
| BART | $2.736 \pm 1.095$ | $0.492 \pm 0.261$ | $6.534 \pm 5.020$ |
| Causal Forest | $2.440 \pm 1.011$ | $0.264 \pm 0.142$ | $7.082 \pm 5.772$ |
| GANITE | $2.471 \pm 1.057$ | $0.821 \pm 0.294$ | $8.124 \pm 5.663$ |
| CEVAE | $2.677 \pm 0.350$ | $0.505 \pm 0.118$ | $2.344 \pm 0.204$ |
| TEDVAE | $1.943 \pm 1.023$ | $0.302 \pm 0.370$ | $5.318 \pm 5.306$ |
| Tarnet | $2.307 \pm 0.586$ | $0.145 \pm 0.139$ | $5.503 \pm 2.432$ |
| CFR | $2.117 \pm 0.433$ | $0.199 \pm 0.115$ | $5.269 \pm 1.858$ |
| PoNet | $\mathbf{1.516 \pm 0.463}$ | $\mathbf{0.114 \pm 0.045}$ | $\mathbf{2.750 \pm 1.142}$ |

model. We consider a range of hyperparameters that are known to influence the model's performance, including the weight $\alpha$ of mutual information minimization regularizer (MIMR), weight $\beta$ of the reconstruction module and the weight $\gamma$ of the confounder balancing module.

To systematically investigate their effects, we design a set of experiments where each hyperparameter is varied independently while keeping others fixed at their default values. We vary the three hyper-parameters to range in $\{0.0001, 0.001, 0.01, 0.1, 1, 10\}$. We report the hyper-parameter analysis in terms of $\sqrt{\epsilon_{PEHE}}$ and $\epsilon_{MSE}$ on different dimension settings of the post-treatment variables. The results are as shown in Figure 9. The experimental results indicate that the model exhibits low sensitivity to the chosen hyperparameters. Specifically, varying the hyperparameter values within the tested range did not have a significant impact on the model's performance. These findings suggest that the model's robustness allows for a wide range of hyperparameter settings without compromising its performance. Still, the performance of the model is relatively better when these hyperparameters range in $\{0.0001, 0.001, 0.01\}$.

### E.4 REAL-WORLD DATA

#### E.4.1 THE AVERAGE TREATMENT EFFECT UNDER DIFFERENT ESTIMATOR

In this section, we introduce several common estimators for estimating ATE from observed real-world data, namely the Difference-in-Mean Estimator, IPW (inverse propensity weighting) estimator, and Doubly-Robust (DR) estimator, to check how much the difference in terms of ATE and CATE obtained by different estimators is.

The difference-in-Mean Estimator (Bon et al.) for ATE is defined as:

$$\hat{ATE}_{Diff} = \frac{1}{m} \sum_{i:T_i=1} Y_i - \frac{1}{k} \sum_{i:T_i=0} Y_i, \tag{21}$$

where $m$ and $k$ are the number of treated and control group, respectively.

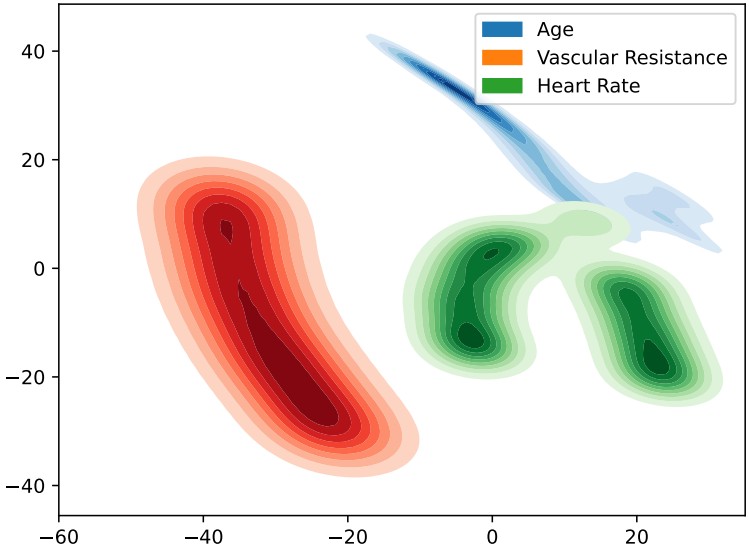

Figure 10: The distribution KDE plot of the representations of three different real-world factors.

The IPW estimator (Bon et al.) for ATE is defined as:

$$\hat{ATE}_{IPW} = \frac{1}{N} \sum_{i=1}^{N} Y_i^*$$
$$with \quad Y_i = Y_i(\frac{T_i}{w_i} - \frac{1 - T_i}{1 - w_i}), \tag{22}$$

where $w_i$ is the propensity score of sample $i$, which could be calculated by logistic regression, $Y_{i*}$ is an unbiased estimator of the CATE according to (Athey & Imbens, 2015).

The Doubly-Robust (DR) estimator (Bon et al.) is defined as:

$$\hat{ATE}_{DR} = \frac{1}{N} \sum_{i=1}^{N} \phi_i^*,$$
$$with \quad \phi_i^* = \hat{\mu}_1(x_i) - \hat{\mu}_0(x_i) + T\frac{Y_i - \hat{\mu}_1(x_i)}{w(x_i)} - (1 - T)\frac{Y_i - \hat{\mu}_0(x_i)}{1 - w(x_i)}, \tag{23}$$

where $\hat{\mu}_1(x_i)$ and $\hat{\mu}_0(x_i)$ are the preliminary estimations of the response surfaces for the treated and control groups, which can be calculated by any regression model (e.g., we can utilize linear regression to get $\hat{\mu}_1(x_i)$) by training on the samples with $T = 1$), $w(x_i)$ is the propensity score for $x_i$. And the estimator $\phi_i^* = \hat{\mu}_1(x_i) - \hat{\mu}_0(x_i) + T\frac{Y_i - \hat{\mu}_1(x_i)}{w(x_i)} - (1 - T)\frac{Y_i - \hat{\mu}_0(x_i)}{1 - w(x_i)}$ is also proven to be another unbiased estimator for CATE (Athey & Imbens, 2015).

First, we report the ATE estimation on real-world data in two different time steps of the data in Table 9 to check their difference: From the results in the table we can see that in fact the difference in the

Table 9: Estimated ATE under different Estimator on real-world dataset.

| Estimator | $t_1$ | $t_2$ |
|---|---|---|
| Difference-in-Mean | -0.268 | -0.448 |
| IPW Estimator | -0.183 | -0.256 |
| Double Robust Estimator | -0.545 | -0.538 |

estimated ATE between different estimators is still quite significant, even for the so-called unbiased IPW estimator and Double Robust estimator.

Furthermore, regarding the IPW and Double Robust estimator, we know that their definitions allow for the estimation of CATE for each sample. We are also interested in comparing the differences

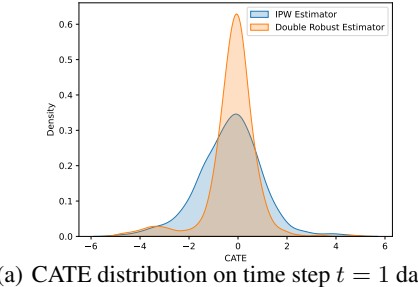

(a) CATE distribution on time step $t = 1$ data

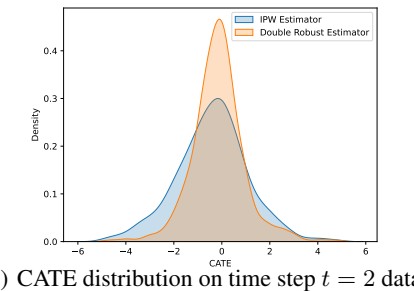

(b) CATE distribution on time step $t = 2$ data

Figure 11: CATE distribution on time step $t = 1$ and $t = 2$ on MIMIC III.

in CATE estimated by these two estimators. Therefore, we recorded the CATE for every sample in real-world data estimated by these two estimators and used kernel density estimation to plot the distribution of CATE estimated by them (fitting with Gaussian distribution). The results are shown in Figure 11. It is evident that even for these two unbiased CATE estimators, there is a noticeable difference in the distribution of estimated CATE for the same group of samples. This analysis informs practitioners in the field of causal inference that when estimating causal effects on real-world data, it is necessary to use different causal effect estimators to verify the effectiveness of the methods from multiple perspectives.

### E.4.2 CASE STUDY

In this study, we present a real-world case study to assess the effectiveness of our proposed model, *PoNet*, in identifying and distinguishing different factors, including confounders, mediation post-treatment variables, and collider post-treatment variables. To demonstrate this, we utilize a real-world case extracted from the MIMIC III dataset. Through a thorough examination of the covariates in the MIMIC III dataset, guided by domain knowledge, we successfully identified the corresponding confounder, mediation, and collider post-treatment variables. Figure 12 illustrates the identified variables in our case study. Specifically, we observed that the patient's systemic vascular resistance is influenced by the use of vasopressors (Treatment), which subsequently impacts the patient's blood pressure (outcome). Consequently, the systemic vascular resistance can be classified as a mediation post-treatment variable. Similarly, we discovered that the patient's heart rate is also affected by the use of vasopressors, and affected by an unmeasured variable health status . However, there is no evidence indicating a direct causal relationship between heart rate and the patient's blood pressure. Hence, heart rate is categorized as a collider post-treatment variable. Additionally, we found that age plays a dual role as it not only influences the likelihood of a patient using vasopressors but also affects blood pressure. Therefore, age is considered a confounder in this context.

Once the *PoNet* model is trained using the training data, we proceed to the inference phase on the test data. During this phase, we selectively mask off the covariates other than age, vascular resistance, and heart rate. This involves setting the values of the remaining covariates to 0, resulting in three separate batches of test data, each containing only one of the aforementioned non-zero covariates. These batches are then fed into the trained *PoNet* model for inference. Then we can obtain the representations of $C$ (confounder), $Z_m$ (mediation post-treatment), and $Z_c$ (collider post-treatment) for the three masked test data sets. Subsequently, we concatenate the representations of $C$, $Z_m$, and $Z_c$ and employe t-SNE to reduce the dimensionality of these representations for each batch. By doing so, we are able to generate a KDE (Kernel Density Estimation) plot that visually depict the relationships between the variables of interest. These plots as shown in Figure 10 provide valuable insights into the distribution of the representations of the covariates under investigation. By the masking operation,

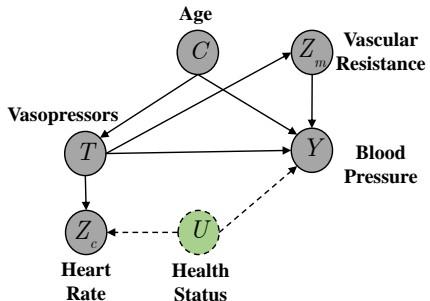

Figure 12: Real-world case causal relationship.

we can see that the distributions of the three real-world covariates are different and separated from each other, which means the three different fators (confounder, mediation post-treatment variable and collider mediation post-treatment variable) can be identified and distinguished by the proposed model *PoNet*.

By thoroughly examining and categorizing these factors, we demonstrate the capability of PoNet in identifying and distinguishing different types of variables in causal analysis. This case study exemplifies the practical relevance of our model and highlights the importance of accurate identification and understanding of confounders, mediation and collider post-treatment variables for successful causal inference.

## F LIMITATIONS

One of the limitations in this work, is that in some extreme scenarios, such as when there is only a single covariate, PoNet might not perform optimally. This limitation arises from PoNet's design, which separates out three distinct types of factors from the covariates. In such extreme cases, our model might not be as effective, but it is important to note that this limitation is not unique to PoNet and applies to other popular causal estimation methods as well. Practitioners should carefully consider the nature of their data and the specific requirements of their research context when choosing a causal estimation method, potentially utilizing alternative approaches as necessary.

Another limitation of this study is our assumption that the mediation and collider post-treatment variables are entirely independent. Our theoretical analysis suggests that to mitigate post-treatment bias, we should condition on mediation post-treatment variables while excluding collider post-treatment variables in the inference policy. However, it is possible for certain post-treatment variables to act as both mediators and colliders simultaneously. This duality introduces the risk of collider post-treatment bias when they are included as conditioning factors in inference, and mediation post-treatment bias when they are not. Consequently, further exploration is warranted in this area. It is crucial to investigate whether there are more theoretically sound strategies to effectively eliminate such biases or to determine the prevalence of dual-type post-treatment variables in real-world settings and carefully select the scenarios for application, thereby offering more insightful guidance to practitioners.