# OpenReview forum: "Extracting Post-Treatment Covariates for Heterogeneous Treatment Effect Estimation"
_ICLR.cc/2024/Conference — Submitted to ICLR 2024_

### Official Review · Reviewer_SsoN · 2023-10-24

**Soundness:** 3 good
**Presentation:** 3 good
**Contribution:** 2 fair
**Rating:** 5
**Confidence:** 4

**Summary:**

This paper studies the heterogeneous treatment effect estimation problem in the setting where post-treatment variables exist in addition to the commonly considered confounders, treatments, and outcomes. As the identity of post-treatment covariates is presumed unknown to the analyst, there could be a source of bias in estimating treatment effects. A deep balancing neural network, PoNet, was proposed to address this issue by incorporating a mutual information regularizer and a reconstruction loss to separate the covariates in latent spaces. Experiments on several datasets show that PoNet achieves better performance than common baselines.

**Strengths:**

1. This paper is clearly written and easy to follow.
2. The experiments are thorough and show a clear advantage over other baselines in the considered setting.

**Weaknesses:**

I have some concerns about the practicality of the considered problem, as well as the proposed method.
1. The assumption that the analyst is unaware of which covariates are confounders and which are post-treatment variables seems quite rare in real-world scenarios. Typically, domain knowledge or prior information can help identify confounders. Therefore, the practical applicability of this assumption in real-world problems may be limited.
2. In cases where the analyst genuinely has no prior knowledge of the causal structure, there are established methods in the causal inference literature for discovering causal relationships from observational data. For example, conditional independence tests can be applied to recover the causal structure, especially since post-treatment variables induce v-structures. This approach may offer a more direct and interpretable solution compared to encouraging conditional independence in an embedding space using mutual information regularizers as in the proposed method. Also, testing conditional independence in the original space seems to be a simpler problem compared to requiring conditional independence in an embedding space.
3. The decoupling approach that combines the reconstruction loss and the mutual information regularizer seems to be a heuristic one since it does not really guarantee that the covariates are properly separated into pre-treatment and post-treatment ones, which could hurt the reliability of the proposed method.

**Questions:**

Please see the Weaknesses part.

---

> ### Author Response · Authors · 2023-11-20
> **Author Response to Reviewer SsoN (1-2)**
>
> # 1.Practical applicability of assumptions in real-world scenarios
>
> Thank you for your valuable insight regarding the assumption in our study. We recognize that in many real-world scenarios, domain knowledge and prior information play a crucial role in identifying confounders and post-treatment variables. We recognize that in many real-world scenarios, domain knowledge and prior information play a crucial role in identifying confounders. However, our assumption is grounded in the reality that there are numerous complex scenarios where such clear distinctions are not readily apparent. Especially in emerging fields or interdisciplinary research, the intricate interplay between various factors might not be fully understood. For example,  in the field of investigating the role of microbiota in Chronic Diseases, particularly gut microbiota, the exact role and mechanisms through which microbiota influence chronic diseases like obesity, diabetes, and heart disease are still under investigation, with many conflicting studies and theories. In the psychological field, there is ongoing debate about which psychotherapy methods are most effective for certain disorders and why they work. Different schools of thought (e.g., cognitive-behavioral, psychodynamic) offer differing explanations and approaches, and the field is constantly evolving with new research.  In such cases, the ability to differentiate between confounders and post-treatment variables can be significantly challenging. This is particularly true in large-scale observational studies where the volume and variety of data can obscure the underlying causal structure.
>
> Moreover, our approach is designed to provide a robust framework for scenarios where domain knowledge is limited or where the causal relationships among variables are not well understood. By allowing for the possibility that the nature of covariates is not fully known, our model offers a potential and general solution against potential post-treatment biases that might arise from the observational data. This can be particularly valuable in exploratory analyses or in fields where causal relationships are still being investigated.
>
> In summary, while we acknowledge that domain knowledge is invaluable in identifying confounders in many situations, our approach caters to those complex and less understood scenarios where such clarity is not available. The flexibility of our model to adapt to both well-understood and ambiguous situations enhances its practical applicability across a wide range of real-world problems.
>
> # 2. Established methods for discovering causal relationships might be more direct
>
> Thank you for your insightful question regarding the use of causal discovery in the context of unknown causal structures in observational data. While it is true that conditional independence tests are a valuable tool in causal discovery, particularly in identifying v-structures induced by post-treatment variables, they also come with certain limitations, especially in complex data environments. For instance, when dealing with high-dimensional covariates, the computational cost of these methods can be quite substantial. Moreover, causal discovery based on conditional independence tests faces the challenge of Markov equivalence classes. This issue means that for some causal structures, it is not possible to determine the direction of causality solely through independence tests. Additionally, many of these methods tend to be linear and may struggle to model causal relationships in high-dimensional and complex environments effectively.
>
> Our proposed method, PoNet, offers an alternative approach by focusing on identifying and distinguishing the distributions of three different types of covariates – confounders, collider post-treatment variables, and mediation post-treatment variables. As we demonstrate in Theorem 1, if we can successfully identify and separate these distributions from the covariates, the causal effect can be identified in the presence of post-treatment variables, without the need to fully understand the specific causal relationships between covariates.
>
> But we deeply appreciate the value of your suggestion regarding the use of causal discovery. Leveraging causal discovery in the estimation of causal effects without predetermined causal structures represents an exciting and promising direction for future exploration in this field.Thank you once again for your insightful suggestion.

---

> ### Author Response · Authors · 2023-11-20
> **Author Response to Reviewer SsoN (3)**
>
> # 3. The heuristic nature of the decoupling approach in the proposed method
>
> Thank you for your critical observation regarding our decoupling approach. We understand your concern about the potential heuristic nature of this approach. However, Our approach in PoNet is grounded in a hypothesized causal graph, from which we establish an inference policy. This policy is then operationalized using a data-driven model to decouple the covariates into three factors: confounders and two types of post-treatment variables. The particular structure of our model is designed as multi-channels based on the treatment assignment, to ensure that the learned representation of confounders is pre-treatment, while the other two factors are post-treatment. Moreover, through the supervised information from the data label and the backpropagation during training, we further ensure that the inferred confounders and mediation post-treatment variables are predictive  to outcome while collider post-treatment variables are irrelevant to the outcome. Then the reconstruction loss ensures that the learned representations retain the essential information from the original covariates, while the MIMR helps in encouraging the independence of the representations of confounders, mediation post-treatment, and collider post-treatment variables. This method aims to balance the need for information preservation and the separation of different types of variables, which is a challenging task in high-dimensional and complex datasets. Moreover, our extensive experimentation, detailed in the paper, demonstrates the effectiveness of our proposed model in evaluation. PoNet consistently outperforms state-of-the-art models across various scenarios, indicating its robustness and reliability in handling post-treatment bias. This empirical evidence supports the efficacy of our method, suggesting that, despite potential theoretical concerns, PoNet is a reliable tool for addressing post-treatment bias in practical applications.

---

> ### Author Response · Authors · 2023-11-23
>
> Dear Reviewer,
>
> I hope this message finds you well. I am writing to follow up on the response we recently submitted addressing the concerns you raised regarding our manuscript, for which you had assigned a score of 5. This score, while moderately favorable, indicates that there were aspects of our work that could be improved.
>
> In response to your valuable feedback, as well as that of other reviewers, we have undertaken significant revisions to our manuscript. These revisions include extensive additional experiments and new theoretical analyses, which have substantially strengthened the overall quality and contribution of our work. We believe that these enhancements have addressed the key areas of concern and have brought our paper more in line with the high standards of the conference. Your expertise and feedback are highly valued, and we would greatly appreciate your reassessment of our revised submission.
>
> Thank you for your consideration, and we look forward to any further insights or feedback you may have.
>
> Best regards,
>
> Authors

---

### Official Review · Reviewer_zkzj · 2023-10-26

**Soundness:** 3 good
**Presentation:** 3 good
**Contribution:** 3 good
**Rating:** 6
**Confidence:** 4

**Summary:**

This work proposes a new method for dealing with post-treatment covariates in treatment effect estimation. Briefly speaking, post-treatment covariates should not be included when estimating causal effects; however, due to the challenging in discerning post vs. pre-treatment covariates and satisfying the unconfoundedness assumption, post-treatment covariates are often present in observational causal inference. The method discuss post-treatment covariates in two scenarios: mediation and collider biases, and provided a solution using feed-forward neural networks. The network structure is inspired by the classic TARNet, which employs two branches for treated and untreated subjects. Experiments are conducted on a selection of synthetic, semi-synthetic, and real-world datasets.

**Strengths:**

1. The idea and motivation for addressing post-treatment covariate bias is novel and important.
2. The quality and clarity of writing is quite good. The paper is easy to follow and the logic is clearly articulated.
3. The selection of baselines is adequate in the experiments.

**Weaknesses:**

1. The technical contribution is somewhat limited. The proposed algorithm feels like a band-aid solution with a structure very similar to TARNet (in terms of using two branches to handle the post-treatment covariates). Based on the experimental results reported in the manuscript, it seems that a generative approach with variable decomposition (that accommodate post-treatment covariates and confounding) can further improve the performance significantly.

**Questions:**

1. What would be the performance of PoNet on datasets without post-treatment covariates? As in real-world scenarios practitioners may not be able to judge if all included covariates are all pre-treatment or not, it is useful if PoNet can be used in both scenarios and achieve state-of-the-art results.

2. In Figure 1(b), the collider case of post-treatment bias, would a variable decomposition CATE estimator (e.g., TEDVAE) be able to address this scenario? This also relates to the MIMIC-3 dataset and may explain why TEDVAE performs the best among all the other compared baselines.

3. Would a generative approach that accommodates variable decomposition and post-treatment covariates have better performance? PoNet is essentially based on modified TARNet,which seems to have moderate performance.

---

> ### Author Response · Authors · 2023-11-20
> **Author Response to Reviewer zkzj (1-2)**
>
> # 1.Limited technical contribution and similarity to existing methods.
>
> Thank you for your comments regarding the technical aspects of our proposed algorithm. We acknowledge that in terms of the model architecture, we have not made extensive alterations from the TARNet framework, because architecture design is not our focus. However, we would like to emphasize that the structure design of the model is not the primary contribution of our work. Our significant contribution lies in unifying two different sources of post-treatment bias within a single framework and distinguishing between confounders and two types of post-treatment variables, which is rarely considered in the field of treatment effect estimation from observational data. In addition, our work importantly advances the theoretical understanding of causal effect identifiability under these complex conditions. We have rigorously proved that, given certain causal assumptions, causal effects are identifiable even in the presence of both confounders and two distinct types of post-treatment variables. Besides, in the current version we also enrich this work's theoretical contribution by proving the inferred confounders and post-treatment factors are minimally sufficient (It refers to a statistic that captures all the information in a sample that is relevant for estimating a parameter of the underlying probability distribution.) for the needed parameter $\theta$ for the true treatment effect, more details can be found in Appendix C.
>
> # 2. Performance of PoNet on datasets without post-treatment covariates.
>
> Thank you for raising the question about the applicability and performance of PoNet in scenarios where datasets may not have post-treatment covariates. In response to your suggestion, we have conducted additional experiments using a new semi-synthetic dataset generated from PeerRead, where post-treatment bias has been removed. The performance on error of PEHE, ATE and MSE on the test data is shown in the following table, we also have updated it in the Appendix:
>
> |                     | $\sqrt{\epsilon_{PEHE}}$ | $\epsilon_{ATE}$ | $\epsilon_{MSE}$ |
> |---------------------|--------------------------|------------------|------------------|
> | LR                  | 2.718 $\pm$ 1.076        | 0.206 $\pm$ 0.089| 6.647 $\pm$ 4.495|
> | OLS2                | 2.279 $\pm$ 0.852        | 0.102 $\pm$ 0.113| 6.403 $\pm$ 4.352|
> | BART                | 2.736 $\pm$ 1.095        | 0.492 $\pm$ 0.261| 6.534 $\pm$ 5.020|
> | Causal Forest       | 2.440 $\pm$ 1.011        | 0.264 $\pm$ 0.142| 7.082 $\pm$ 5.772|
> | GANITE              | 2.471 $\pm$ 1.057        | 0.821 $\pm$ 0.294| 8.124 $\pm$ 5.663|
> | CEVAE               | 2.677 $\pm$ 0.350        | 0.505 $\pm$ 0.118| 2.344 $\pm$ 0.204|
> | TEDVAE              | 1.943 $\pm$ 1.023        | 0.302 $\pm$ 0.370| 5.318 $\pm$ 5.306|
> | Tarnet              | 2.307 $\pm$ 0.586        | 0.145 $\pm$ 0.139| 5.503 $\pm$ 2.432|
> | CFR                 | 2.117 $\pm$ 0.433        | 0.199 $\pm$ 0.115| 5.269 $\pm$ 1.858|
> | PoNet               | **1.516 $\pm$ 0.463**    | **0.114 $\pm$ 0.045**| **2.750 $\pm$ 1.142**|
>
> The results indicate that PoNet is not only adaptable to this situation but also maintains its efficiency, often achieving state-of-the-art results compared to other methods. Essentially, if the dataset only contains pre-treatment covariates, PoNet will focus on adjusting for  confounders to estimate the treatment effect accurately .Furthermore, our model’s  inferred​ $Z_c$,  which is excluded to predict outcome, in the absence of post-treatment bias, essentially acts to remove noise variables from the original covariates. This feature contributes to PoNet's superior performance in this scenario. The results from these experiments on the PeerRead dataset without post-treatment variables demonstrate that PoNet can effectively handle both post-treatment and confounding biases, showcasing its robustness and versatility.

---

> ### Author Response · Authors · 2023-11-20
> **Author Response to Reviewer zkzj (3-4)**
>
> # 3. Applicability of a variable decomposition CATE estimator in the collider case.
>
> Thank you for your question. As analyzed in our paper, in the scenario of collider post-treatment bias, if an estimator does not separate collider post-treatment variables from the original covariates and excludes these variables from the outcome estimation, collider bias would still persist. This holds true even for decomposition CATE estimators such as TEDVAE. Regarding your observation that TEDVAE performs best among all compared baselines on the real-world dataset, it's important to note that although TEDVAE does not specifically account for post-treatment scenarios, its objective is to decompose three distinct factors: confounders, risk factors (which only affect the outcome), and instrumental factors (which only affect treatment and do not affect the outcome). In the context of collider post-treatment bias, one of our goals is to learn a factor $Z_c$ that does not affect the outcome, which aligns with the instrumental factors in TEDVAE: For the prediction of the outcome, both the collider factor $Z_c$ and the instrumental factor can be considered as noise. Both PoNet and TEDVAE effectively exclude these factors when predicting the outcome. Consequently, this approach may contribute to TEDVAE's superior performance on the real-world dataset compared to other baseline methods. In summary, while TEDVAE and similar estimators do not directly address post-treatment scenarios, their methodological approach to factor decomposition aligns with the objectives of learning non-affecting factors, which can inadvertently contribute to their effective performance in the presence of post-treatment bias.
>
> # 4. The potential of a generative approach accommodating variable decomposition and post-treatment covariates.
>
> Thank you for your thought-provoking question regarding the potential of a generative approach to address post-treatment bias. We acknowledge the merit in considering a generative approach for handling post-treatment bias. Indeed, generative models, which establish joint distributions between input and output data, can offer comprehensive insights into data relationships, potentially providing a robust framework for addressing complex issues like post-treatment bias. However, it's important to recognize that generative models come with their own set of assumptions and requirements, particularly regarding the distribution of data. They necessitate assumptions and modeling of the data distribution, which can be challenging for complex datasets, especially when computational resources are limited. The strong assumptions required for generative models may not always be suitable for smaller-scale datasets or for those with intricate distribution characteristics. Despite these challenges, we agree that exploring post-treatment bias from a generative perspective is a promising avenue. This approach could potentially offer novel insights and solutions to this pervasive issue in causal inference. We are grateful for your suggestion and intend to further explore this direction in our future work. This exploration would contribute to the ongoing efforts in the field to develop more versatile and effective methods for addressing various biases in causal inference.

---

### Official Review · Reviewer_mav9 · 2023-11-01

**Soundness:** 2 fair
**Presentation:** 1 poor
**Contribution:** 1 poor
**Rating:** 5
**Confidence:** 3

**Summary:**

The paper discusses post-treatment bias in estimating Conditional Average Treatment Effects (CATEs). It provides a decomposition for this bias under linearity and also discuss identifiability. A neural model is then proposed for estimating the latent structure of the covariates and using this to adjust estimated CATEs.

**Strengths:**

See below for a contextual discussion of strengths and perceived weaknesses.

**Weaknesses:**

The paper is right to note that post-treatment bias can pose problems for causal identification. Conditioning on more information doesn't always improve ATE or CATE estimation. Distinguishing types of variables as pre- or post-treatment is a valuable, albeit difficult, goal.

This discussion in mind, there are a few factors serving as perceived weaknesses.

The paper as currently presented spends some time and effort on establishing the well-known problem of post-treatment basis as a problem (e.g., see Abstract and Section 2.2). Some of this is certainly useful to remind readers of the problem as motivation, but the issue is well-known so less emphasis on that would help highlight the contribution here more squarely. Then, Theorem 1 could be better connected to the post-treatment bias problem, with the key point seeming to be that, under a certain causal structure, knowledge of the conditional distribution of pure confounders and post-treatment variables given all the variables

There are also some aspects to the paper that seem to distract from the main point. For example, the discussion of MIMR is framed as a contribution of the paper ("We propose a Mutual Information Minimization Regularizer (MIMR)"), although I think it could be somewhat further emphasized what problem the regularization is trying to solve (I appreciate the empirical results on the MIMR, however).

Another thought concerns use of the proposed DAG. Post-treatment variables are presumably omnipresent, and it can be difficult to know how to proceed when they are around (motivating this paper). For applicability in practice, I do not know when or whether investigators would be willing to assume the proposed graph in Figure 1c. Knowing more about the limitations of the proposed approach would help (e.g., discussion of limitations [the paper does not seem to currently have a limitations section]). For example, if I just have a single covariate, and I don't know whether it is a pre- or post-treatment variable, the proposed method would not work (at least according to my understanding of the paper). How many covariates one "needs", and information like that, would be helpful to contextualize the strengths and limits of the contribution.

Overall, the selected rating balances (a) the importance of the problem faced, but (b) perceived limitations with overall clarity of presentation, as well as perceived practical limitations regarding the method's use.

A few detailed points:

The text ''separation of confounders" should read "Separation of confounders" on p. 5.

**Questions:**

In the proposed DAG, I see that X -> C. If, for example, X is simply partitioned deterministically into the two Z and then the one C component, would this imply that p(C|X) is a degenerate distribution? In other words, if the "C" part of X is, say, age, then p(age | {age, Z}) would be either 0 or 1? If so, would this be a problem? Or am I thinking about this in the wrong way? (Theorem 1 uses p(C|X)

---

> ### Author Response · Authors · 2023-11-20
> **Author Response to Reviewer mav9 (1-2)**
>
> # 1.Some redundant discussion of well-known issues and the connection of Theorem 1
>
> We appreciate your suggestion to reduce the emphasis on the general problem of post-treatment bias, given its established recognition in the field. We agree that a more focused approach would indeed allow us to highlight the unique contributions of our work more effectively. But detailing the two sources of different post-treatment bias in the paper is crucial for the readability of the article and for the reader to understand the contribution of our work. Based on your suggestions, we have tried to shorten the description of the post-treatment bias in the revised version.
>
> Regarding Theorem 1, we acknowledge the need for a clearer connection to the specific problem of post-treatment bias. Our intention with Theorem 1 is to establish that under certain causal structures, the knowledge of the conditional distribution of pure confounders and post-treatment variables given all variables is critical for identifying causal effect. In Theorem 1, we demonstrate a key insight: under the given conditions of covariates $X$ and treatment assignment $T$, the probability distribution of an outcome under an intervention $T$ is determined by the distribution of confounders and mediation post-treatment variables, rather than by collider post-treatment variables. This aligns directly with our analysis presented in Section 2.2. In that section, we discuss the importance of including mediation post-treatment variables in causal effect estimation to capture the indirect effects of treatment on outcome through the mediation variable $Z_m$. Conversely, we argue against conditioning on collider post-treatment variables $Z_c$, as this could introduce additional bias due to unobserved variables. This aligns perfectly with the conclusion of Theorem 1, which underscores the importance of accurately identifying and including the right variables in causal models to avoid post-treatment bias. And this theorem is pivotal in demonstrating how our proposed method effectively handles the complexities introduced by post-treatment variables, thereby mitigating post-treatment bias. Our work with Theorem 1 thus directly addresses the challenge of post-treatment bias by elucidating the specific types of variables that should be considered in the causal modeling process. This theoretical grounding strengthens the basis for our proposed model, PoNet, which is designed to effectively distinguish and utilize the appropriate post-treatment variables for accurate causal inference. Thank you for highlighting the need to more explicitly connect Theorem 1 to the issue of post-treatment bias. Your feedback has provided us with an opportunity to clarify this crucial aspect of our paper. We have clarified it in the updated version.
>
> # 2.Some distracting aspects, e.g., the discussion of MIMR.
>
> We appreciate your appreciation of the empirical results regarding MIMR, and we agree that a more focused discussion on the specific problem it addresses would enhance the paper's clarity.
>
> The primary purpose of the MIMR, as detailed in Section 3.3 of our paper, is to ensure the accurate separation of confounders and post-treatment variables, which is essential for unbiased treatment effect estimation. In situations where representations of confounders include information from mediation post-treatment variables, controlling for confounders might inadvertently introduce mediation post-treatment bias. Similarly, if mediation post-treatment variables contain confounder information, addressing confounding bias might not be fully effective.  if  mediation post-treatment variables $Z_m$ contain collider post-treatment information, conditioning on $Z_m$ to predict the outcome can lead to collider post-treatment bias as illustrated in Section 2.2. Precise differentiation between confounders and post-treatment variables is thus critical for reliable treatment effect estimation. To address this challenge, we utilize the MIMR based on kernel density estimation. This non-parametric method fits the distributions of the representations of these variables and measures their independence. By minimizing the mutual information between these variable representations, we effectively separate the underlying factors from each other, thus preventing the inadvertent introduction of bias into the treatment effect estimation. This separation is critical, as it allows for more precise differentiation between confounders and post-treatment variables, ensuring reliable treatment effect estimation. The MIMR thus plays a pivotal role in enhancing the robustness and accuracy of PoNet, our proposed model, by ensuring the independence of critical variables in the causal inference process. We hope this explanation clarifies the specific problem that MIMR addresses and its importance in the context of our work. We have emphasized this aspect more clearly in the revised version of our paper.

---

> ### Author Response · Authors · 2023-11-20
> **Author Response to Reviewer mav9 (3-4)**
>
> # 3. Concerns about DAG,and limitations of the proposed approach
>
> Thank you for your insightful query about graph structure as depicted in Figure 1c. Our study acknowledges and addresses the omnipresence of post-treatment variables in real-world scenarios and the complexities they introduce in causal inference. The proposed DAG in our paper is designed to illustrate a **general** framework for handling these complexities, but we understand that its applicability might vary based on the specific context and knowledge of the covariates involved.
>
> The question of how many covariates are "needed" for the effective application of the proposed model is also an important consideration. Our experiments, which encompassed synthetic, semi-synthetic, real world datasets with dimensions ranging from tens to thousands of covariates, have demonstrated that PoNet is effective even in scenarios where the number of covariates is relatively small. This effectiveness is particularly notable in comparison to methods that only consider confounders. The strength of PoNet lies in its data-driven decomposition to identify and distinguish between pre-treatment and post-treatment covariates when the pre-post nature of covariates is unknown. We acknowledge, however, that in some extreme scenarios, such as when there is only a single covariate, PoNet might not perform optimally. This limitation arises from PoNet's design, which separates out three distinct types of factors from the covariates. In such extreme cases, our model might not be as effective, but it is important to note that this limitation is not unique to PoNet and applies to other popular causal estimation methods as well. We appreciate your insightful perspective on this matter. It highlights an interesting point about the relationship between the choice of estimation method and the contextual background knowledge of the application scenario. Practitioners should carefully consider the nature of their data and the specific requirements of their research context when choosing a causal estimation method.
>
> Another limitation of this study is our assumption that the mediation and collider post-treatment variables are entirely independent. Our theoretical analysis suggests that to mitigate post-treatment bias, we should condition on mediation post-treatment variables while excluding collider post-treatment variables in the inference policy. However, it is possible for certain post-treatment variables to act as both mediators and colliders simultaneously. This duality introduces the risk of collider post-treatment bias when they are included as conditioning factors in inference, and mediation post-treatment bias when they are not. Consequently, further exploration is warranted in this area. It is crucial to investigate whether there are more theoretically sound strategies to effectively eliminate such biases or to determine the prevalence of dual-type post-treatment variables in real-world settings and carefully select the scenarios for application, thereby offering more insightful guidance to practitioners.
>
> We have updated the above limitation discussion in the updated Appendix. Thank you very much for your valuable comments!
> # 4. Question about the DAG in terms of degenerate distribution
>
> In our DAG, the arrow $X \rightarrow C, Z_c, Z_m$ indicates that the confounders $C$ and post-treatment variables $Z_m, Z_c$ are a subset or a function of the covariates $X$. This does not necessarily imply a deterministic partitioning of $x$ into distinct components $Z_m$, $Z_c$ and $C$. Instead, it suggests that $C$ (and $Z_m$, $Z_c$) can be derived or inferred from $X$. We just use expression forms such as  $P(C|X)$ to highlight the distributions of the factors of interest can be inferred from covariate.
>
> But we do understand your confusion and concerns regarding this expression form, in response, we have followed the work of Zhang et al 2021 to provide a more simplified proof of Theorem 1. This re-formulated proof also hinges on Theorem 1 that if we can infer the distributions of $C$, $Z_m$ and $Z_c$, then the causal effect is identifiable under the assumed causal structure. The detailed proof, which we have now updated in the appendix, is as follows:
>
> $P(Y_u|do(T),X)
> =P(Y|do(T),C,Z_c,Z_m,U)\\
> =P(Y|do(T),C,Z_m,U)\\
> =P(Y|T,C,Z_m,U)\\
> =P(Y_u|T,C,Z_m)
> $
>
> This proof, although reformulated, employs the same tools and principles as the original expression form, grounded in the causal graph we proposed, the corresponding manipulated graph, and the three rules of the do-calculus. The last term of the derivation is precisely the same as $P(Y_u|T,Z_m,C)$ in the last term of our original formal derivation (See Appendix C), but here the process of inferring Z_c, Z_m and C from X is omitted. We believe this clarified approach more effectively addresses your concern and elucidates how we apply these theoretical tools to support the causal inference capabilities of our model.

---

> ### Author Response · Authors · 2023-11-23
> **Gentle  inquiry for Reviewer Feedback**
>
> Dear Reviewer,
>
> I hope this message finds you well. I am writing to inquire about the response we recently submitted addressing your concerns regarding our manuscript. We have not yet received your feedback, and we are eager to know if our responses and revisions have adequately addressed your concerns.
>
> We deeply appreciate the score of 3 you initially assigned, which we believe reflects your honest and critical evaluation of our work. In light of this, we have taken your feedback very seriously. Following the insights provided by you and other reviewers, we have conducted substantial additional experiments and developed new theoretical analyses. These efforts have led to solid modifications and significant improvements in our manuscript. Understanding the rigorous standards of the review process, we have worked diligently to enhance the quality and impact of our paper. We believe that these revisions have substantially addressed the initial concerns and have enriched the contribution of our work.
>
> Considering the extent of these revisions and enhancements, we kindly request you to reconsider the evaluation of our paper. We would be grateful if you could review the updated version of our manuscript and provide your valuable assessment.
>
> We understand the demands on your time and appreciate your dedication to maintaining the high standards of the conference. Your expertise and feedback are crucial to us, and we sincerely hope that our revised submission aligns more closely with these standards.
>
> Thank you very much for your time and consideration. We look forward to your thoughts and are hopeful for a positive re-evaluation.
>
> Best regards,
>
> Authors

---

### Official Review · Reviewer_XTaX · 2023-11-01

**Soundness:** 3 good
**Presentation:** 3 good
**Contribution:** 3 good
**Rating:** 6
**Confidence:** 4

**Summary:**

In the article "Extracting Post-Treatment Covariates for Heterogeneous Treatment Effect Estimation", the authors propose to address the problem of post-treatment bias. Based on the literature that suggests to learn a relevant representation from observed variables to perform better adjustment in causal inference, this work introduces a new decomposition of the observed variables to distinguish three types of covariates: confounders, colliders and mediators, of which only confounders should be adjusted for. The authors provide a theoretical proof of the identifiability of the estimand under a new set of assumptions. Finally, the paper presents several experiments to illustrate the performances f the proposed approach, PoNet, on simulated and real data.

**Strengths:**

The article is well written, and quite clear. The problem is clearly exposed, and the proposed solution is quite elegant.

**Weaknesses:**

1. The figures and tables legends are not to be read on their own. It is a very nice feature to be able to understand the figures without reading the main text (at least for variables definitions, etc). Additionally, the text on figures is generally too small, in particular for Figure 2 which is barely readable.

2. the source code is not provided. The absence of a ready-to-use code for practitioners is a severe limit to the usefulness of this work by other researchers in the community, and mostly practitioners.

3. The theoretical part is a little limited, it could be enriched for instance with a bound on the reconstruction from the obtained representations.

**Questions:**

4. can you include the method proposed by Li et al. 2022 in the benchmark? as you mention it partially solves the problem of post-treatment bias you address.

5. it would be interesting to consider the error on the ATE, in addition to the CATE, as an evaluation metric.

6. regarding the experiment on real data, is it possible to also report the different CATE (and ATE) estimates? To have an idea of the impact of the different methods on the actual estimand of interest, even if we do not know the true value.

7. can you provide more guidance for the practitioner? It would be valuable to suggest if PoNet should be applied in all cases, including when post-treatment bias is not really an issue, i.e. extend the experiments to show the performance of PoNet compared to existing approaches in the absence of post-treatment bias. It would illustrate the performance of PoNet on the sole problem of confounder adjustment. Additionally, the performances of PoNet on other datasets used in the original publications of the other methods in the benchmark could be interesting to report (and a very nice contribution to the evaluation standards for the causal representation learning community), though the workload is important.

8. can you report some details regarding the applicability of PoNet? in particular running times, requirement of a GPU or a CPU? variation of the resource need depending on the number of observations and their dimension, as well as the variation of performances depending on those numbers, i.e. does one need a sizable dataset to use PoNet?

9. a detail, the first sentence of section 3.3 is incomplete

---

> ### Author Response · Authors · 2023-11-20
> **Author Response to reviewer XTaX (1-4)**
>
> # 1.Legends in figures and tables are not clear enough.
>
> Thank you for pointing this out. We are sorry that the legends in the figures and tables are not clear for reading. In the revised version, we have revised all figures and tables to include comprehensive legends. These will detail all variables and key aspects, ensuring they can be understood independently of the main text. We appreciate your attention to these details.
>
> # 2. Source code is not provided.
>
> Thank you for your insightful feedback regarding the availability of the source code for our work. We would like to kindly point out that the source code for our model is indeed available and the link has been provided in the supplementary material. Specifically, the link to the code can be found in Appendix Section E, which discusses additional experiments. We understand how crucial it is for the code to be easily discoverable and accessible, and we apologize if its location in the supplementary material was not immediately apparent. To address this, we have put the code link at the beginning of the experiment section in the main body of the paper.
>
> # 3. The theoretical part could be more enriched.
>
> Thank you for your valuable feedback regarding the theoretical contributions of our paper. We appreciate your suggestion to enrich the theoretical framework.
>
> Firstly, we would like to highlight that our paper already proposes a significant theoretical contribution regarding the identification of causal treatment in the presence of both confounders and post-treatment variables, as detailed in Theorem 1. We have provided extensive proof for this in the appendix.
>
> Regarding the bounds of reconstruction loss, we have thoroughly considered this aspect. The reconstruction loss based on mean square error, a common regularization term in deep learning, serves dual purposes in our work. It ensures learning about collider post-treatment variables during backpropagation and retains the information in the original covariates. However, the challenge lies in the fact that the reconstructed covariates are processed through a nonlinear decoder based on a multi-layer perceptron. This creates a non-convex problem for solving the bound of this reconstruction loss, presenting a complex challenge in optimization.
>
> But we still totally understand your concerns about the theoretical depth, thus we have further enriched our paper from another angle in the revised version. Noting that Theorem 1 indicates that the treatment effect can be identified by recovering $C$ and $Z_m$, we further provide a new theorem:
>
> The joint set of inferred factors for $C$, $Z_m$ is minimally sufficient for the optimal parameters $\theta$ which estimation of unbiased treatment effects needs.
>
>  It refers to the inferred confounders and mediation post-treatment underlying factors are minimally sufficient for recovering the true treatment effect's optimal parameters. This theory asserts that the inferred post-treatment variables and confounders are the most parsimonious statistics, yet they retain all necessary information for estimating the optimal parameters required for a sufficient estimation of the true treatment effect. The theory has been updated in Section 2 of the main body and detailed proof of this new theorem is also provided in the appendix C.
>
> # 4. Inclusion of a method proposed by Li et al. 2022 in the benchmark.
>
> Thank you for your suggestion to include the method proposed by Li et al. 2022 in our benchmark comparisons. We acknowledge the relevance of this method, as it addresses one aspect of post-treatment bias, which is a key focus of our work. we indeed intended to include Li et al. 2022's method in our comparative analysis. However,  the source code for their method is not publicly available. The absence of this code posed a barrier to implementing their method accurately for a fair and rigorous comparison in our experiments. we are open to including this comparison in future work, should the source code become available or if we can collaborate with the authors for an accurate implementation. In the meantime, we have made efforts to ensure that our benchmark comparisons are comprehensive and include a range of relevant methods that are available for comparison. We appreciate your understanding in this matter and are grateful for your suggestion.

---

> ### Author Response · Authors · 2023-11-20
> **Author Response to reviewer XTaX (5-6)**
>
> # 5. Consideration of error on ATE in addition to CATE.
>
> We are immensely grateful for your constructive suggestion to consider the error on ATE, in addition to the CATE, as an evaluation metric in our study. Acting on your valuable feedback, we have re-conducted our experiments to include the error metrics of ATE on the synthetic and semi-synthetic datasets in our evaluation.  For example, the updated results regarding to the ATE metric on semi-synthetic PeerRead  are as follows:
>
> |    PeerRead   |        d=50        |        d=100       |       d=200       |
> |:-------------:|:------------------:|:------------------:|:-----------------:|
> |       LR      |  0.405 $\pm$ 0.164 | 0.317 $\pm$ 0.343  | 1.140 $\pm$ 0.785 |
> |      OLS2     | 0.147 $\pm$ 0.067  | 0.363 $\pm$ 0.074  | 0.723 $\pm$ 0.108 |
> |      BART     | 0.209 $\pm$ 0.089  | 0.445 $\pm$ 0.241  | 1.253 $\pm$ 0.613 |
> | Causal Forest | 0.172 $\pm$ 0.066  | 0.386 $\pm$ 0.183  | 0.938 $\pm$ 0.390 |
> |     CEVAE     | 0.415$ \pm$ 0.079  | 0.544 $\pm$ 0.125  | 0.761$\pm$0.357   |
> |     GANITE    | 1.740 $\pm$ 0.390  | 1.748 $\pm$ 0.243  | 0.506 $\pm$ 0.396 |
> |     TEDVAE    | 0.339 $\pm$ 0.169  | 0.399 $\pm$ 0.314  | 0.599 $\pm$ 0.639 |
> |     Tarnet    | 1.018 $\pm$ 0.210  | 0.498 $\pm$ 0.198  | 0.757 $\pm$ 0.246 |
> |      CFR      | 1.045 $\pm$ 0.181  |  0.544 $\pm$ 0.220 | 0.706 $\pm$ 0.233 |
> |      PoNet     | 0.125 $\pm$ 0.058  | 0.131 $\pm$ 0.077  | 0.138 $\pm$ 0.137 |
>
> The results of these additional experiments are now incorporated in our paper. Specifically, the results on the synthetic and  semi-synthetic datasets are presented in Table 6 and 7 in Appendix. Encouragingly, these additional experimental results demonstrate that our proposed method continues to outperform other baseline methods in terms of the ATE error metrics. This further validates the effectiveness of our approach and strengthens our contributions to the field.
>
> # 6. Report of CATE and ATE estimates on real data.
>
> Thank you for your question about reporting the CATE  and ATE estimates on real-world data.
>
> Regarding CATE estimates, due to the nature of real-world data, which lacks counterfactual outcomes for each sample, it is unfortunately not feasible to calculate CATE as it is defined by the average root mean square difference between the actual and predicted CATE for each sample: ${\epsilon_{PEHE}}=\sqrt{\frac{1}{N}\sum_{i=1}(\tau_i-\hat{\tau}_i)^2}$,
>
> where  $\tau_i=y_{i}^{t_{i}=1}-y_{i}^{t_{i}=0}, \hat{\tau_i}= \hat{y}_i^{t_i=1}-\hat{y}_i^{t_i=0}$ are the ground truth CATE and the estimated CATE, respectively.
>
> For the ATE error metric, while the standard definition involves the absolute difference between the true ATE and the predicted ATE: $ |Error_{ate}| = |ATE -\hat{ATE}| = \left| \frac{1}{n} \sum_{i=1}^{n}(y_i(1) - y_i(0)) - \sum_{i=1}^{n}(\hat{y}_i(1) - \hat{y}_i(0)) \right|$, the actual calculation requires knowledge of each sample's true counterfactual outcome, which is not available in real-world data.
>
>  However, understanding the importance of your inquiry, we approximated the ATE calculation as follows:  $ATE = \frac{1}{m}\sum_{i=1}^{m} y_i(1) - \frac{1}{k}\sum_{i=1}^{k} y_i(0)$, where m and k are the number of samples in treated and control group. Then we conducted experiments on real-world data using this approximation. In our study, we focused on our proposed method PoNet, along with CFR and TarNet (excluding others like OLS1, OLS2, TEDVAE, BART, Causal Forest, GANITE, CEVAE due to their significantly larger error values). We recorded the ATE error metrics for these methods every 10 epochs during 100 training epochs:
>
> |        | epoch=1 | epoch=10 | epoch=20 | epoch=30 | epoch=40 | epoch=50 | epoch=60 | epoch=70 | epoch=80 | epoch=100 |
> |:------:|:-------:|:--------:|:--------:|:--------:|:--------:|:--------:|:--------:|:--------:|:--------:|:---------:|
> |  PoNet |  0.023  |   0.18   |   0.153  |   0.071  |   0.03   |   0.051  |   0.016  |   0.033  |   0.098  |   0.151   |
> |   CFR  |  0.282  |   0.095  |   0.028  |   0.088  |   0.096  |   0.092  |   0.117  |   0.345  |   0.14   |   0.062   |
> | TARnet |  0.346  |   0.121  |   0.015  |   0.076  |   0.084  |   0.095  |   0.263  |   0.187  |   0.059  |   0.033   |
>
> The results  indicate that the ATE error metrics are highly unstable, appearing somewhat random. This instability is likely due to the unavailability of true ATE values in real-world data, suggesting that calculating ATE error metrics without counterfactual outcomes may not be a reliable evaluation metric in real-world settings. We hope this explanation clarifies our approach and findings, and we appreciate your understanding of the inherent limitations when working with real-world data.

---

> ### Author Response · Authors · 2023-11-20
> **Author Response to reviewer XTaX (7-9)**
>
> # 7. More guidance for practitioners and extending experiments.
>
> Thank you for your insightful suggestion to provide more guidance for practitioners and to extend our experiments to scenarios where post-treatment bias is not a significant issue. In response to your suggestion, we have conducted additional experiments using a new semi-synthetic dataset generated from PeerRead, where post-treatment bias has been removed. The performance on error of PEHE, ATE and MSE on the test data is shown in the following table, we also have updated it in Table 8  in the Appendix:
>
> |                     | $\sqrt{\epsilon_{PEHE}}$ | $\epsilon_{ATE}$ | $\epsilon_{MSE}$ |
> |---------------------|--------------------------|------------------|------------------|
> | LR                  | 2.718 $\pm$ 1.076        | 0.206 $\pm$ 0.089| 6.647 $\pm$ 4.495|
> | OLS2                | 2.279 $\pm$ 0.852        | 0.102 $\pm$ 0.113| 6.403 $\pm$ 4.352|
> | BART                | 2.736 $\pm$ 1.095        | 0.492 $\pm$ 0.261| 6.534 $\pm$ 5.020|
> | Causal Forest       | 2.440 $\pm$ 1.011        | 0.264 $\pm$ 0.142| 7.082 $\pm$ 5.772|
> | GANITE              | 2.471 $\pm$ 1.057        | 0.821 $\pm$ 0.294| 8.124 $\pm$ 5.663|
> | CEVAE               | 2.677 $\pm$ 0.350        | 0.505 $\pm$ 0.118| 2.344 $\pm$ 0.204|
> | TEDVAE              | 1.943 $\pm$ 1.023        | 0.302 $\pm$ 0.370| 5.318 $\pm$ 5.306|
> | Tarnet              | 2.307 $\pm$ 0.586        | 0.145 $\pm$ 0.139| 5.503 $\pm$ 2.432|
> | CFR                 | 2.117 $\pm$ 0.433        | 0.199 $\pm$ 0.115| 5.269 $\pm$ 1.858|
> | PoNet               | **1.516 $\pm$ 0.463**    | **0.114 $\pm$ 0.045**| **2.750 $\pm$ 1.142**|
>
>  These experiments have been enlightening, showing that PoNet is adept not only at eliminating post-treatment bias but also at addressing confounding bias,which is achieved through the balancing module in PoNet, which adjusts for the confounders .Furthermore, our model inferred​ $Z_c$,  which is excluded to  predict the outcome, in the absence of post-treatment bias, essentially acts to remove noise variables from the original covariates. This feature contributes to PoNet's superior performance in this scenario. The results from these experiments on the PeerRead dataset demonstrate that PoNet can effectively handle both post-treatment and confounding biases, showcasing its robustness and versatility.
>
> While we fully acknowledge and support your idea of comparing the performances of different methods on standard benchmark datasets without post-treatment bias, due to time constraints and the extensive workload required, we plan to undertake this additional experimentation in future work. However, we believe the results from our extended experiments on the PeerRead dataset already provide evidence of PoNet's efficacy in various scenarios to a certain extent. We appreciate your suggestion and are committed to further exploring these avenues to enhance the utility and applicability of PoNet in the broader causal inference landscape.
>
> # 8. Applicability details of PoNet (running times, resource needs).
>
> Thank you for your inquiry regarding the practical applicability details of PoNet, including running times and hardware requirements. For the experiments conducted in our study, we utilized an Intel(R) Xeon(R) Gold 5120 CPU @ 2.20GHz with 512GB of RAM, and an NVIDIA TITAN RTX GPU with 24 GB of memory. The datasets used in our experiments varied in size, with the synthetic dataset having dimensions (15000, 24/48/72), the semi-synthetic dataset (7601, 1020/1153/1344), and the real-world dataset (6133, 69). Focusing on the computational aspects, taking the largest semi-synthetic dataset used in our experiments with size (7601, 1344) as an example, the PoNet model exhibited efficient performance. On average, each epoch of training took approximately 1.1 seconds without batch partitioning. The peak GPU memory consumption was around 1500MB.
>
> As for the requirement of sizable datasets, our experiments indicate that PoNet performs effectively on the datasets of the sizes we tested. We believe PoNet to be adaptable to various dataset sizes, although the optimal performance is subject to the complexity and nature of the data. We hope this information assists practitioners in understanding the applicability and resource requirements of PoNet and facilitates its adoption in relevant fields.
>
> # 9. about  the first sentence of section 3.3 is incomplete
>
> Thank you for pointing this out. The sentence should be: Separating confounders and post-treatment variables is essential for unbiased treatment effect estimation. We have corrected it in the revised version.

---

> > ### Comment · Reviewer_XTaX · 2023-11-22
> > **response to the authors**
> >
> > The authors have addressed most of my remarks in a satisfying way, with encouraging results!
> >
> > There has just been a misunderstanding regarding point number 6. The idea was simply to report the value each method finds for the ATE and the CATE (a function rather than a single value of course). Indeed, in synthetic data, one often designs the data to obtain differences between the methods, and it would be interesting to check if the resulting effect estimations are very different, and not only evaluate the prediction of y.
> >
> > It is indeed very difficult to have metrics for causal effects, as the true effect is unknown. I am not sure about the metric you suggest, it seems to be subject to confounding bias?

---

> > > ### Author Response · Authors · 2023-11-23
> > > **Further discussion on ponit number 6**
> > >
> > > Dear reviewer,
> > >
> > > Thank you for your clarification regarding point number 6. We appreciate this clarification and recognize the importance to report the value different method finds for the ATE and the CATE.
> > >
> > > As you mentioned, it is indeed very difficult to have metrics for causal effects, as the true effect is unknown. But in light of your suggestion, we use several common estimators for estimating ATE from observed real-world data, namely the Difference-in-Mean Estimator, IPW (inverse propensity weighting) estimator, and Doubly-Robust (DR) estimator, to check how much the difference in terms of ATE and CATE obtained by different estimators is.
> > >
> > > The difference-in-Mean Estimator  for ATE is defined as:
> > >
> > > $\hat{ATE}\_{Diff}=\frac{1}{m}\sum_{i:T_i=1}Y_i-\frac{1}{k}\sum_{i:T_i=0}Y_i$, where $m$ and $k$ are the number of treated and control group, respectively.
> > >
> > > The IPW estimator  for ATE is defined as:
> > >
> > > $\hat{ATE}\_{IPW}=\frac{1}{N}\sum_{i=1}^NY_i^* \\
> > > with \quad Y_i=Y_i(\frac{T_i}{w_i}-\frac{1-T_i}{1-w_i})$, where $w_i$ is the propensity score of sample $i$, which could be calculated by logistic regression, $Y_i*$ is an unbiased estimator of the CATE according to Athey et.al 2015.
> > >
> > > The Doubly-Robust (DR) estimator is defined as:
> > >
> > > $\hat{ATE}\_{DR}=\frac{1}{N}\sum_{i=1}^{N}\phi_i^*,
> > > with \quad \phi\_i^*=\hat{\mu}\_1(x_i)-\hat{\mu}_0(x_i)+T\frac{Y_i-\hat{\mu}_1(x_i)}{w(x_i)}-(1-T)\frac{Y_i-\hat{\mu}_0(x_i)}{1-w(x_i)}$,where $\hat{\mu}_1(x_i)$ and $\hat{\mu}_0(x_i)$ are the preliminary estimations of the response surfaces for the treated and control groups, which can be calculated by any regression model (e.g., we can utilize linear regression to get $\hat{\mu}_1(x_i))$ by training on the samples with $T=1$), $w(x_i)$ is the propensity score for $x_i$. And the estimator $\phi_i^*=\hat{\mu}_1(x_i)-\hat{\mu}_0(x_i)+T\frac{Y_i-\hat{\mu}_1(x_i)}{w(x_i)}-(1-T)\frac{Y_i-\hat{\mu}_0(x_i)}{1-w(x_i)}$ is also proven to be another unbiased estimator for CATE (Athey et.al 2015).
> > >
> > > First, we report the ATE estimation using the above estimators on the real-world data in two different time steps of the data in the following Table:
> > >
> > > |         Estimator        |  $t_1$ |  $t_2$ |
> > > |:------------------------:|:------:|:------:|
> > > |    Difference-in-Mean    | -0.268 | -0.448 |
> > > |       IPW Estimator      | -0.183 | -0.256 |
> > > | Double Robust Estimator  | -0.545 | -0.538 |
> > >
> > > From the results in the table we can see that in fact the difference in the estimated ATE between different estimators is still quite significant, even for the so-called unbiased IPW estimator and Double Robust estimator.
> > >
> > > Furthermore, regarding the IPW and Double Robust estimator, we know that their definitions allow for the estimation of CATE for each sample. Following your feedback, we are further interested in comparing the differences in CATE estimated by these two estimators. Therefore, we recorded the CATE for every sample in real-world data estimated by these two estimators and used kernel density estimation to plot the distribution of CATE estimated by them (fitting with Gaussian distribution). The results are shown in **Figure 11** in the Appendix. It is evident that even for these two unbiased CATE estimators, there is a noticeable difference in the distribution of estimated CATE for the same group of samples.  This analysis informs practitioners in the field of causal inference that when estimating causal effects on real-world data, it is necessary to use different causal effect estimators to verify the effectiveness of the methods from multiple perspectives.
> > >
> > > We are grateful for your feedback and the opportunity to enhance the depth and breadth of our analysis. We believe that incorporating this analysis will significantly enrich our study and provide a more comprehensive evaluation. And we have updated the above contents in the revised version of our submission.

---

> > > ### Author Response · Authors · 2023-11-23
> > > **On  suggested metric is subject to confounding bias**
> > >
> > > Dear reviewer,
> > > Thank you for your insightful comments on the suggested metric. As you can see in my last response, the suggested metric in my response (6) is indeed based on the Difference-in-Mean esitmator for ATE.
> > >
> > > The difference-in-means estimator is only considered an unbiased estimator for the average treatment effect under certain conditions. The conditions for its unbiasedness typically include:
> > >
> > > 1.  **Random Assignment**: The treatment is randomly assigned to the subjects. This ensures that both the treatment and control groups are statistically equivalent in expectation, which is crucial for unbiasedness.
> > >
> > > 2. **No Selection Bias**: There is no selection bias in the sample. This means that the sample of subjects in the study is representative of the population for which the treatment effect is to be estimated.
> > >
> > > 3. **No Attrition or Non-compliance**: There is no attrition (i.e., loss of participants) or non-compliance (participants not adhering to the assigned treatment or control condition) that is systematically related to the treatment. Such issues, if correlated with the treatment, can introduce bias.
> > >
> > > 4. **SUTVA (Stable Unit Treatment Value Assumption)**: The potential outcomes for any unit (subject) are unaffected by the treatments assigned to other units. This means there should be no interference between units or hidden variations in the treatment.
> > >
> > > Due to the presence of confounders, the above Difference-in-Mean esitmator-based metric clearly does not satisfy all of the above conditions, and therefore, as you said, it suffers from confounding bias, which is why I analyzed in my last response that it may not be a reliable metric in the treatment effect estimation task. We are grateful for your question and hope this explanation addresses your concerns about this metric.

---

### Author Response · Authors · 2023-11-23

Dear Program Chairs, Senior Area Chairs, Area Chairs and Reviewers,

I hope this message finds you well. I am writing to express my sincere gratitude for the opportunity to submit our work to ICLR and for the valuable feedback provided by the reviewers.

In response to the invaluable feedback, we have conducted additional experiments, enriched the theoretical contribution and provided more comprehensive anlysis, and thoroughly revised our paper. These efforts were aimed at enhancing the clarity, robustness, and overall quality of our research. The revisions and efforts we made can be summarized by the following:

**(1)** First, We have revised parts of our manuscript at the writing level in response to the reviewers' suggestions, aiming to enhance the readability and clarify our contributions.

**(2)** Second, we have enriched the theoretical contribution of our article; we proved that the inferred confounders and mediation post-treatment variables are statistically minimally sufficient for the parameters necessary for recoverying  treatment effect.

**(3)** Third, we have included additional experimental results on synthetic and semi-synthetic datasets, comparing our proposed method with other baselines in terms of ATE error metrics. The results showcase our model's strong performance in group-level treatment effect estimation.

**(4)** Fourth, we also conducted experiments to evaluate the performance of our proposed model and other baselines in scenarios without post-treatment bias. These results highlight the robustness and versatility of our model.

**(5)** Fifth, we have carried out an analysis of treatment effect evaluation on real-world data, specifically addressing point 6 and subsequent follow-ups to reviewer XTaX. We believe this analysis significantly enriches our study and offers a more comprehensive evaluation, providing valuable insights for practitioners of causal inference when assessing real-world data.

We understand and respect the rigorous standards of the conference and the review process. We are grateful for the opportunity to improve our work based on the constructive feedback provided by the reviewers. We kindly request that the reviewers and committee consider the revisions and additional work undertaken in response to the reviewers' feedback. We hope that these efforts might positively influence the ratings of our submission and final evaluation.

Thank you once again for your time and consideration. We look forward to the possibility of contributing to the ICLR community.

Best,

Authors

---

### Meta-Review · Area_Chair_DvEC · 2023-12-06

**Metareview:**

The authors study a setting where  covariates can be affected by the treatment and collected post-treatment. They propose a causal graph (Figure 1.c) to handle post-treatment bias. However, the justification for using this specific graph is not fully convincing. In particular, the independence assumptions inherent in this graph are critical for the approach to work, yet the validity of these assumptions is not defended thoroughly. Given that the entire analysis hinges on this structure, it would be necessary to either provide stronger justification for this choice of causal structure, or discuss methods to test t​he implications of it.

Apart from Theorem 1, which adapts the G-formula to the proposed causal graph, the paper lacks significant theoretical contributions. This gap is pronounced when compared to similar works on heterogeneous treatment effect estimation, some of which the authors cite, which provide theoretical guarantees for their approaches. The validity of PoNEt approach is solely backed by empirical evidence on synthetic and semi-synthetic datasets, which favor their approach, specially by generating independent post-treatment covariates.

I have also a concern regarding the theorem added in the revision, Theorem 2. Unfortunately neither the statement, nor the proof of thsi theorem is clear.  The proof of this theorem consists of proving rather trivial statements which are irrelevant to the main claim of the theorem. Also, the lemmas attributed to Silvey 2017 seem to be either basic definitions or results known in statistics for over a century (e.g., Fisher 1922, etc.)

To conclude, the authors are considering an interesting and important problem in causal inference, which can potentially inspire follow up research. The of adapting representation learning to the case of separating pre- and post-treatment covariates, although limited to the graph of Figure 1.c, seems novel and inspiring. Yet given all the issues regarding Theorem 2 and the current state of the paper, I cannot recommend accept. I believe the paper could be significantly improved with a well thought-out rather than a rushed revision.

**Justification For Why Not Higher Score:**

Please see the meta review and issues pertaining to the assumptions as well as the theoretical results.

**Justification For Why Not Lower Score:**

The authors study an interesting problem.

---

### Decision · Program_Chairs · 2024-01-16

Reject